# Beyond *Leishmania*: hidden trypanosomatid diversity reveals complex parasite-sand fly networks in southeastern Brazil

Ana Paula Isnard, Amanda Caroline Corrêa Madureira, Gustavo Mayr de Lima Carvalho, Mariana Lourenço Freire, Daniel Moreira de Avelar, Lileia Gonçalves Diotaiuti, José Dilermando Andrade Filho, Felipe Dutra-Rêgo/+

Fundação Oswaldo Cruz-Fiocruz, Instituto René Rachou, Belo Horizonte, MG, Brasil

**BACKGROUND** Sand flies (Diptera: Psychodidae) are well-known vectors of *Leishmania*, yet their associations with other trypanosomatids remain poorly understood. Expanding knowledge on these interactions is essential to elucidate the ecological diversity of parasites circulating in natural and periurban environments.

**OBJECTIVE** To characterise sand fly species composition and assess the diversity of trypanosomatids naturally infecting sand flies in the Serra do Cipó district, Minas Gerais, Brazil.

**METHODS** Sand flies were collected between 2023 and 2025 in the Mata da Tapera and nine surrounding peridomestic sites using Centre for Disease Control light trap (CDC-LT) and a Shannon trap. Females were examined by midgut dissection and screened individually for trypanosomatids using nested polymerase chain reaction (PCR) targeting the 18S rRNA V7-V8 region, followed by sequencing and phylogenetic analyses.

**FINDINGS** A total of 1,460 sand flies representing 21 species were collected, with *Pintomyia pessoai* (35.5%) being the most abundant. No flagellates were observed in 105 dissected females. Molecular screening of 730 females revealed 12 positives (overall positivity = 1.6%), including *Leishmania infantum* in *Pi. pessoai*, *Pi. christenseni*, and *Pa. barretoi*; *L. braziliensis* in *Pi. monticola*; and non-*Leishmania* taxa such as *Herpetomonas samuelpessoai*, *Novymonas esmeraldas*, a representative of Strigomonadinae, *Trypanosoma* sp. (Anura clade), and a lineage related to *Sergeia*.

**MAIN CONCLUSION** These findings confirm the circulation of *Leishmania* in Serra do Cipó while revealing hidden trypanosomatid diversity spanning at least five genera. The results suggest that sand flies may act as ecological "hubs", transiently interacting with multiple parasite lineages beyond the classical *Leishmania* cycle, highlighting the need to broaden the ecological perspective in sand fly-parasite studies.

Key words: Phlebotominae - *Leishmania* - Trypanosomatidae - phylogenetic analysis - vector-parasite interactions

Sand flies (Diptera: Psychodidae: Phlebotominae) are highly diverse insects widely distributed in tropical and subtropical regions.[1] Their medical relevance is primarily linked to their role as vectors of *Leishmania* (Kinetoplastea: Trypanosomatidae), the causative agents of leishmaniasis, a disease that remains a major public health challenge in Brazil and many other countries.[2] Both cutaneous leishmaniasis (CL) and visceral leishmaniasis (VL) are endemic in Minas Gerais State, where periurban transmission has become increasingly frequent and has expanded into areas until recently considered free of the disease.[3,4,5]

Although the association between sand flies and *Leishmania* is well established, other trypanosomatids are increasingly reported in these insects, particularly in Brazil, including *Endotrypanum*,[6,7] *Herpetomonas*,[8] *Crithidia*,[8,9] and *Trypanosoma*.[10] These findings suggest that sand flies may interact with a broader range of monoxenous and dixenous parasites than traditionally recognised. While the epidemiological significance of these associations remains uncertain, their detection complicates the interpretation of molecular results in targeted surveillance of *Leishmania*, especially when using broad-range markers that can amplify multiple trypanosomatid lineages.

Periurban environments, shaped by habitat fragmentation, human settlement, and overlapping wild and domestic reservoirs, provide a unique scenario for exploring these interactions.[11] In such contexts, shifts in sand fly composition and abundance may influence not only classical *Leishmania* transmission cycles but also the occurrence of other trypanosomatids.[12] For *Leishmania*, it is well established that species naturally suited to exploit anthropised environments, such as *Lutzomyia longipalpis*, *Nyssomyia whitmani*, and *Migonemyia migonei*, play a central role in sustaining transmission in these areas.[13,14,15,16,17] In contrast, whether sand flies that thrive in urban or periurban settings also contribute to the per-

**doi:** 10.1590/0074-02760250260

+ **Corresponding author:** felipedutra04@hotmail.com | https://orcid.org/0000-0002-2799-8267

**Handling editor:** Elisa Cupolillo | https://orcid.org/0000-0002-0620-3250

sistence or dispersal of other trypanosomatids, or merely reflect incidental associations, remains unclear. This uncertainty limits our understanding of parasite-vector interactions in anthropised landscapes.

Here, we address this gap by investigating sand flies from a forest fragment and peridomestic settings in Serra do Cipó, Minas Gerais, Brazil, to explore their species composition and to assess the occurrence of trypanosomatids. This region is part of one of Brazil's major biodiversity hotspots and has previously reported the presence of sand fly vectors[18] as well as autochthonous cases of canine visceral leishmaniasis (Municipal Health Department of Santana do Riacho).

## MATERIALS AND METHODS

*Study area and sand fly collection* - The study was conducted in the Serra do Cipó district (19º20′10.97″S, 43º37′48.75″W), municipality of Santana do Riacho, Minas Gerais, Brazil. This region forms part of the United Nations Educational, Scientific and Cultural Organisation (UNESCO)-designated Serra do Espinhaço Biosphere Reserve, a heterogeneous landscape that integrates Atlantic Forest and Cerrado elements and includes several conservation units, among them the Serra do Cipó National Park.

Sand fly collections focused primarily on the Mata da Tapera forest fragment (MT) (19º19′50.84″S, 43º36′57.03″W) where five sampling campaigns were conducted between March 2023 and July 2024. Twenty Centre for Disease Control light trap (CDC-LT) were installed approximately 1 m above ground and operated from 6:00 p.m. to 6:00 a.m. for three consecutive nights in each campaign. In July 2024, a single Shannon trap collection was also performed from 6:00 p.m. to 9:00 p.m.

In addition, nine peridomestic sites located in the immediate surroundings of the forest fragment were surveyed once as a preliminary assessment of sand fly fauna at the forest-household interface. Because only one campaign was conducted in peridomestic areas, these samples were not intended for direct ecological comparison with MT but rather to document species presence and explore potential parasite-vector interactions beyond the forest interior. The spatial distribution of all sampling sites is shown in Fig. 1.

Live sand flies collected in CDC-LT were removed from the cages using a Castro aspirator, whereas specimens from the Shannon trap were aspirated directly from the capture cloth. All live individuals were kept in transport boxes under controlled conditions [25ºC, 80% relative humidity (RH)] and were dissected for midgut exam-

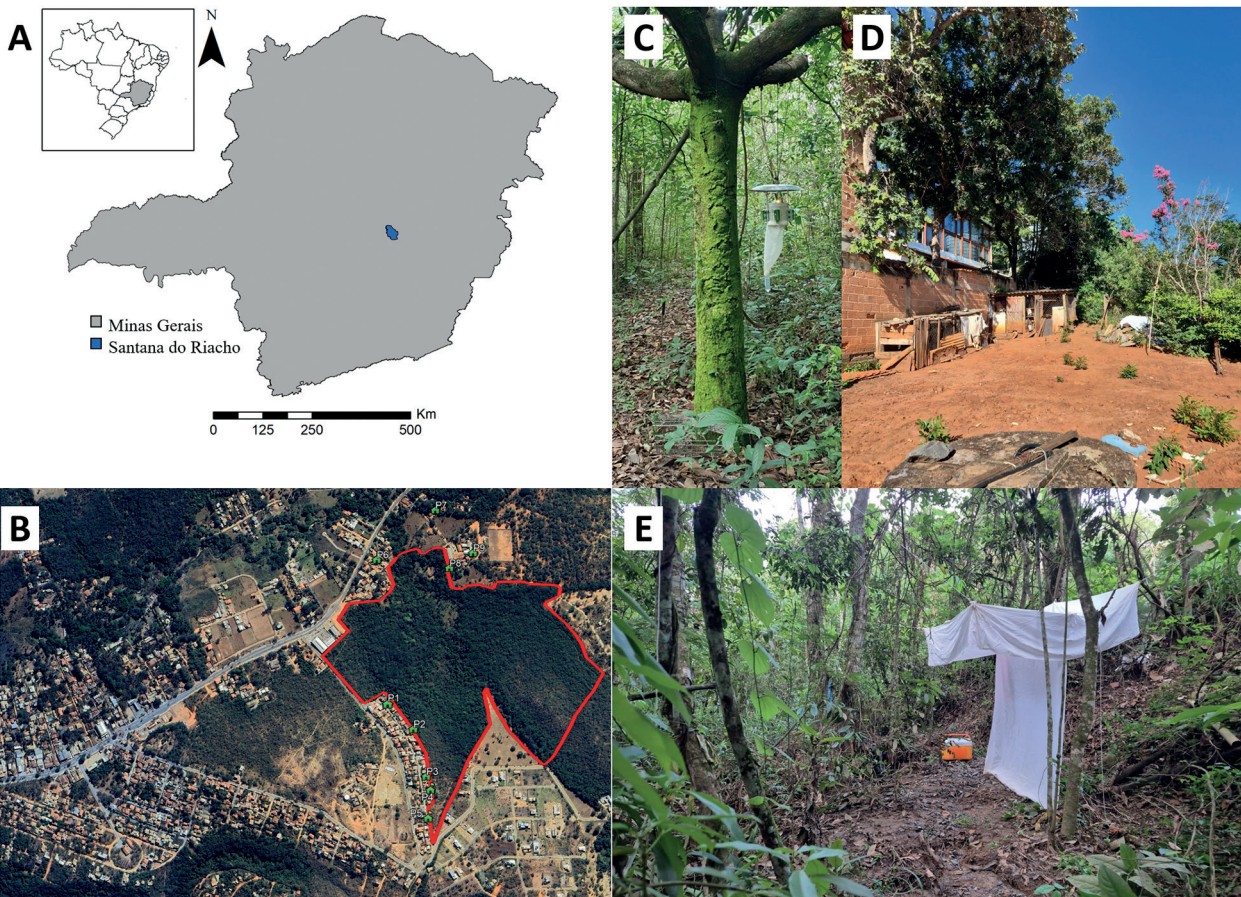

Fig. 1: study area and sand fly sampling sites. (A) Location of Serra do Cipó district (Santana do Riacho, Minas Gerais, Brazil). (B) Mata da Tapera (MT) forest fragment (red polygon) and the nine peridomestic sites surveyed in this study. (C) Distribution of Centre for Disease Control light trap (CDC-LT) within the MT. (D) Peridomestic sampling sites surrounding the forest fragment where CDC-LT were deployed. (E) Shannon trap collection conducted in July 2024 at the MT.

ination. The remaining specimens were preserved in 70% ethanol for subsequent morphological identification and molecular screening. Dissected individuals were not included in the polymerase chain reaction (PCR) analyses.

For the peridomestic survey, nine sites were selected in January 2025 based on their location at the border of MT, as well as the presence of domestic animals and backyard vegetation. At each site, two CDC-LT were operated for three consecutive nights under the same conditions as described above. All sand flies from peridomestic collections were preserved in 70% ethanol.

*Natural infection of Trypanosomatidae* - Live females were transported to Fiocruz Minas (Belo Horizonte, Brazil) for midgut dissection on glass slides. Specimens were first chilled at -20ºC for 5 min and then washed three times in a 1:1 solution of distilled water and commercial detergent to remove bristles. After washing, the insects were maintained in sterile 1× phosphate-buffered saline (PBS) until dissection. The presence of promastigote forms in the gut was examined under light microscopy at 400× magnification, and the stage of the gonotrophic cycle was determined by analysing the ovaries and accessory glands.[19,20]

*Molecular detection of trypanosomatids* - Females not selected for midgut examination were nonetheless dissected individually for taxonomic identification. Each specimen was dissected with sterile needles on a slide containing 1× PBS, separating the head and the last three abdominal segments for morphological identification.[21] The remaining body parts (thorax and abdomen) were stored separately in dry 1.5 mL tubes at -20ºC for subsequent molecular screening. Generic abbreviations follow the nomenclatural recommendations of Marcondes.[22]

DNA was extracted using the Gentra Puregene kit (Qiagen) following the manufacturer's protocol. Extraction blanks (kit reagents without biological material) were included as negative controls. DNA purity and concentration were measured spectrophotometrically (NanoDrop One, Thermo Scientific), and the integrity of sand fly DNA was verified by PCR amplification of the *cytochrome c oxidase subunit I* (*COI*) gene using primers LCO1490 (5′-GGTCAACAAATCATAAAGATATTGG-3′) and HCO2198 (5′-TAAACTTCAGGGTGACCAAAAAATCA-3′) under previously described conditions.[23] Reactions were performed in 25 µL containing 1× buffer, 2.5 mM MgCl₂, 0.2 mM dNTPs, 0.1 µM of each primer and 2 U Taq DNA. Cycling conditions were: 95ºC for 2 min; 35 cycles of 95ºC for 60 s, 54ºC for 60 s and 72ºC for 90 s; and a final extension at 72ºC for 10 min. For each PCR run, a no-template control (NTC) was included, consisting of the full reaction mix in which the sample DNA was replaced by nuclease-free water.

Detection of trypanosomatids was performed by nested PCR targeting the V7-V8 region of the *18S rRNA* gene, using primer pairs SSU561F (5′-TGGGATAACAAAGGAGCA-3′) / SSU561R (5′-CTGAGACTGTAACCTCAAAGC-3′) in the first reaction, and TRY927F (5′-CAGAAACGAAACACGGGAG-3′) / TRY927R (5′-CCTACTGGGCAGCTTGGA-3′) in the second.[24,25] Reactions were performed in 25 µL containing 1× buffer,

2.0 mM MgCl₂, 0.2 mM dNTPs, 0.32 µM of each primer and 1 U Taq polymerase. First-round amplification used: 94ºC for 5 min; 30 cycles of 94ºC for 30 s, 55ºC for 60 s and 72ºC for 90 s; final extension at 72ºC for 5 min. The products of the first-round reaction using primers were diluted 1:25 in sterile water and 5 µL was used as template in the second round under the same conditions. Each reaction included *Leishmania braziliensis* DNA (MHOM/BH/1975/M2903) as a positive control and a NTC. Male sand flies were also processed as negative controls for both extraction and amplification.

PCR products were purified with the QIAquick PCR Purification Kit (Qiagen) and sequenced bidirectionally by the Sanger method. Forward and reverse chromatograms were inspected manually in FinchTV, and consensus sequences were assembled considering only bases with a minimum Phred quality score of 30. Ambiguous sites, when present, were evaluated by checking the chromatogram peak profiles and resolved by adopting the appropriate International Union of Pure and Applied Chemistry (IUPAC) ambiguity code when necessary. Primer sequences were trimmed from both ends prior to downstream analyses. The resulting consensus sequences were compared against GenBank using BLAST and subsequently used for phylogenetic inference.

*Phylogenetic analyses* - Sequences generated in this study were analysed together with reference sequences retrieved from the GenBank database. Reference sequences were selected by BLAST searches targeting the subfamilies Leishmaniinae, Phytomonadinae, Strigomonadinae and Trypanosomatinae, with *Bodo saltans* included as the outgroup. Sequences were aligned using the employing information from nocal structural interactions (E-INS-i) algorithm implemented in MAFFT v7.505.[26] The V7-V8 alignment was subsequently refined by automated trimming with trimAl v1.5 using a gap-threshold of 0.5.[27] Maximum likelihood (ML) inference was performed in IQ-TREE v2.4.0[28] under the best-fit substitution model TIM3+F+I+G4 selected by ModelFinder.[29] Node support was assessed with 1,000 ultrafast bootstrap replicates and 1,000 SH-aLRT tests.[30] Bayesian inference was performed in MrBayes v3.2.7 under the GTR+F+I+G model for 5,000,000 generations, sampling every 100, with remaining parameters set to default.[31]

*Ecological analysis and interaction matrix* - To evaluate sampling completeness and estimate species richness, rarefaction and extrapolation analyses using the iNterpolation and EXTrapolation (iNEXT) framework were applied.[32,33] Analyses were based on abundance data, considering Hill numbers for $q = 0$ (species richness), $q = 1$ (Shannon diversity, exponential form), and $q = 2$ (Simpson diversity). Sample-size and coverage-based curves were generated with 2,000 bootstrap replications to estimate 95% confidence intervals (CI). *Chao*1 richness was also computed (EstimateS v9.1.0) for comparison with observed values.

To visualise vector-parasite associations, the Sankey diagram was generated in Flourish (https://flourish.studio) using the 'arbitrary flows' mode with automatic node ordering ('Reduce overlaps') to minimise line crossings.

*Ethical statements* - Sand fly collections were approved by the Authorisation and Information on Biodiversity System (SISBIO; permit number 86644-1). The study was registered in the National System for the Management of Genetic Heritage and Associated Traditional Knowledge (SisGen) under registration number A47F256. Collection performed at peridomestic settings were approved by the owners (anonymised here) sampling site locations are summarised in the Supplementary data (Table I).

### RESULTS

A total of 1,460 sand flies were collected during the study, with 1,226 from the MT (84%) and 234 from peridomestic areas (16%), representing 21 species across nine genera. *Pi. pessoai* was the most abundant species (35.5%), followed by *Pi. monticola* (23.6%) and *Ev. evandroi* (7.8%) (Table I). No engorged females were obtained in any of the sampling campaigns, which prevented blood-meal source analyses.

Among the specimens collected in MT, 663 were females (54%) and 563 were males (46%). *Pi. pessoai* was the predominant species (N = 494; 40.2%), followed by *Pi. monticola* (N = 338; 27.5%) and *Pi. christenseni* (N = 81; 6.6%) [Supplementary data (Table II)]. In the Shannon collection conducted in July 2024, 52 sand flies were captured: 36 males, all identified as *Pi. pessoai*, and 16 females, comprising *Pi. pessoai* (N = 8; 50%), *Pi. monticola* (N = 4; 25%), *Mi. quinquefer* (N = 2; 12.5%), *Ny. whitmani* (N = 1; 6.3%), and *Pa. aragaoi* (N = 1; 6.3%).

The species accumulation curve stabilised from the fourth CDC collection campaign onward, indicating that sampling was sufficient to capture local richness (21 species). This was corroborated by the *Chao*1 estimator [*Chao*1 = 21; standard deviation (SD) = 0.12]. Rarefaction and extrapolation curves further indicated that additional sampling would contribute little to the observed diversity, given that the 95% CI of the extrapolated richness overlapped with the observed estimate

TABLE I

Sand flies collected with Centre for Disease Control light trap (CDC-LT) and a Shannon trap between 2023 and 2025 in the Mata da Tapera (MT) and in peridomestic settings surrounding MT in the Serra do Cipó district, Minas Gerais, Brazil

| Sand fly | MT | | Peridomestic settings | | Total (%) |
|---|---|---|---|---|---|
| | ♂ | ♀ | ♂ | ♀ | |
| *Brumptomyia brumpti* | 6 | 5 | 1 | 1 | 13 (0.9) |
| *Evandromyia bacula* | 0 | 3 | 0 | 0 | 3 (0.2) |
| *Evandromyia cortelezzii* | 11 | 17 | 23 | 18 | 69 (4.7) |
| *Evandromyia evandroi* | 20 | 23 | 39 | 32 | 114 (7.8) |
| *Evandromyia teratodes* | 0 | 2 | 0 | 0 | 2 (0.1) |
| *Evandromyia termitophila* | 1 | 1 | 0 | 0 | 2 (0.1) |
| *Lutzomyia ischnacantha* | 1 | 7 | 0 | 0 | 8 (0.5) |
| *Lutzomyia longipalpis* | 26 | 4 | 54 | 5 | 89 (6.1) |
| *Micropygomyia longipennis* | 2 | 4 | 0 | 0 | 6 (0.4) |
| *Micropygomyia quinquefer* | 0 | 5 | 0 | 0 | 5 (0.3) |
| *Migonemyia migonei* | 2 | 0 | 3 | 1 | 6 (0.4) |
| *Nyssomyia whitmani* | 28 | 22 | 8 | 6 | 64 (4.4) |
| *Pintomyia christenseni* | 29 | 52 | 0 | 7 | 88 (6.0) |
| *Pintomyia monticola* | 88 | 250 | 4 | 2 | 344 (23.6) |
| *Pintomyia pessoai* | 294 | 200 | 19 | 5 | 482 (35.5) |
| *Psathyromyia aragaoi* | 46 | 22 | 0 | 0 | 68 (4.7) |
| *Psathyromyia barretoi* | 0 | 23 | 0 | 0 | 23 (1.6) |
| *Psathyromyia brasiliensis* | 0 | 3 | 0 | 0 | 3 (0.2) |
| *Psathyromyia lutziana* | 3 | 6 | 0 | 0 | 9 (0.6) |
| *Sciopemyia birali* | 0 | 1 | 0 | 1 | 2 (0.1) |
| *Sciopemyia sordellii* | 6 | 13 | 0 | 5 | 24 (1.6) |
| Total (%) | 563 (46) | 663 (54) | 151 (64.5) | 83 (35.5) | 1.460 (100) |
| | 1.226 (84) | | 234 (16) | | |

(Fig. 2). Coverage-based rarefaction and extrapolation curves also indicated high sample completeness (> 95%) [Supplementary data (Figs 1-2)].

Natural infection was assessed by dissecting 105 live females, including 89 collected with CDC-LT and 16 from the Shannon trap (Table II). The most frequently dissected species was *Pi. pessoai* (N = 52; 49.5%), followed by *Pi. monticola* (N = 22; 21%) and *Ev. evandroi* (N = 8; 7.6%). Among the dissected specimens, 34% (N = 36) were classified as nulliparous and 66% (N = 69) as parous. No flagellates were observed in the examined samples.

A total of 647 females collected in MT with CDC-LT were screened for trypanosomatids using nested PCR targeting the V7-V8 region of the *18S rRNA* gene. Ten specimens tested positive: *Pi. monticola* (N = 5), *Pi. pessoai* (N = 2), and one each of *Mi. longipennis*, *Pi. chris-*

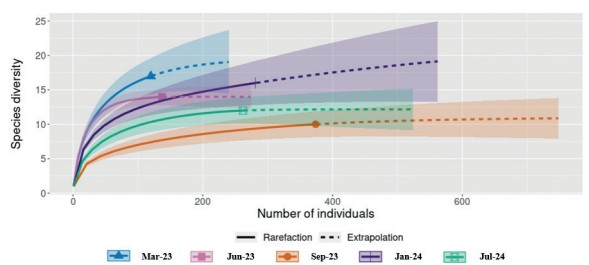

Fig. 2: rarefaction and extrapolation of sand fly species richness in the Mata da Tapera (MT). Rarefaction (solid lines) and extrapolation (dashed lines) curves were generated using the iNterpolation and EXTrapolation (iNEXT) framework based on abundance data, with 2,000 bootstrap replications to estimate 95% confidence intervals (CI) (shaded areas). Species richness stabilised after the fourth collection campaign, with little increase predicted under extrapolation. Observed richness (21 species) coincided with the *Chao*1 estimator [21; standard deviation (SD) = 0.12], indicating that sampling effort was sufficient to capture local diversity.

*tenseni*, and *Pa. barretoi*. Sanger sequencing (Table III) and phylogenetic analysis of the V7-V8 region revealed the presence of *Leishmania infantum* in *Pi. christenseni* (N = 1), *Pi. pessoai* (N = 1), and *Pa. barretoi* (N = 1); *L. braziliensis* in *Pi. monticola* (N = 2); *Herpetomonas samuelpessoai* in *Pi. monticola* and *Pi. pessoai* (N = 1, each); *Trypanosoma* sp. in *Mi. longipennis*; *Novymonas esmeraldas* in *Pi. monticola* (N = 1); and a representative species within Strigomonadinae in *Pi. Monticola* (N = 1). All sequences have been deposited in GenBank under accession numbers PX260227-PX260238. The overall positivity rate for trypanosomatids in MT was 1.5%.

In peridomestic settings, a total of 234 sand flies were collected in January 2025 (males = 151; 64.5%; females = 83; 35.5%). This peridomestic survey represents an initial assessment of the forest-household interface. The most abundant species were *Ev. evandroi* (N = 71; 30.3%), *Lu. longipalpis* (N = 59; 25.2%), and *Ev. cortelezzii* (N = 41; 17.5%) [Supplementary data (Table III)]. All females were screened for trypanosomatids using nested PCR targeting the V7-V8 region of the *18S rRNA* gene, and two specimens tested positive: one *Sc. sordellii* collected at site P4 was positive for *Trypanosoma* sp., and one *Ny. whitmani* collected at site P5 was positive for a trypanosomatid related to the genus *Sergeia* (typically associated with biting midges). The overall positivity rate for trypanosomatids in peridomestic settings was 2.4%.

For all positive samples, ML and Bayesian inference analyses yielded congruent topologies (Figs 3-4, respectively). The interaction matrix between the seven sand fly species that tested positive and the trypanosomatids detected in the study area reveals a complex network ranging from dixenous pathogens of medical relevance, such as *Leishmania* parasites, to less explored lineages, including *Novymonas*, *Trypanosoma* sp. from the Anura clade, and parasites related to *Sergeia* genus. Altogether, at least five subfamilies within Trypanoso-

## TABLE II

Female sand flies collected in the Mata da Tapera (MT), Serra do Cipó district, Minas Gerais, Brazil, and tested for natural *Leishmania* infection

| Sand fly | Mar 23 | Jun 23 | Set 23 | Jan 24 | Jul 24* (CDC/SH) | Total (%) |
|---|---|---|---|---|---|---|
| *Brumptomyia* sp. | - | 3 | - | - | - | 3 (2.9) |
| *Evandromyia cortelezzii* | 3 | - | - | 1 | - | 4 (3.8) |
| *Ev. evandroi* | 1 | - | 6 | 1 | - | 8 (7.6) |
| *Lutzomyia longipalpis* | 3 | - | - | - | - | 3 (2.9) |
| *Micropygomyia quinquefer* | 1 | - | 2 | - | 0 / 2 | 5 (4.8) |
| *Nyssomyia whitmani* | - | - | - | - | 0 / 1 | 1 (1.0) |
| *Pintomyia christenseni* | 1 | - | 4 | - | - | 5 (4.8) |
| *Pi. monticola* | 2 | - | - | 16 | 0 / 4 | 22 (21.0) |
| *Pi. pessoai* | 4 | - | 13 | 14 | 13 / 8 | 52 (49.5) |
| *Psathyromyia aragaoi* | 1 | - | - | - | 0 / 1 | 2 (1.9) |
| Total | 16 | 3 | 25 | 32 | 13 / 16 | 105 (100) |

*In July 2024, live females collected with both Centre for Disease Control light trap (CDC-LT) and a Shannon trap were used for the natural infection survey.

TABLE III

Trypanosomatids in polymerase chain reaction (PCR)-positive female sand flies (Nested PCR targeting the V7-V8 region of the *18S rRNA* gene) collected in the Mata da Tapera (MT) and in peridomestic settings of the Serra do Cipó district, Minas Gerais, Brazil

| Access number (GenBank) | Sand fly | Collection date / site | Trypanosomatidae | Query cover | Identity |
|---|---|---|---|---|---|
| PX260227 | *Pintomyia monticola* | Jan-24 / sylvatic | *Leishmania braziliensis* | 100% | 100% |
| PX260228 | *Pi. monticola* | Jan-24 / sylvatic | *Leishmania braziliensis* | 100% | 100% |
| PX260229 | *Pi. christenseni* | Jun-23 / sylvatic | *Leishmania infantum* | 100% | 100% |
| PX260230 | *Psathyromyia barretoi* | Jun-23 / sylvatic | *Leishmania infantum* | 100% | 100% |
| PX260231 | *Pi. pessoai* | Jul-24 / sylvatic | *Leishmania infantum* | 100% | 100% |
| PX260232 | *Pi. monticola* | Jul-24 / sylvatic | *Novymonas esmeraldas* | 100% | 100% |
| PX260233 | *Pi. monticola* | Jan-24 / sylvatic | *Herpetomonas samuelpessoai* | 100% | 100% |
| PX260234 | *Pi. pessoai* | Jan-24 / sylvatic | *Herpetomonas samuelpessoai* | 100% | 100% |
| PX260235 | *Pi. monticola* | Jan-24 / sylvatic | Trypanosomatidae sp. | 100% | 99.8% |
| PX260236 | *Micropygomyia longipennis* | Mar-23 / sylvatic | *Trypanosoma* sp. | 100% | 100% |
| PX260237 | *Nyssomyia whitmani* | Jan-25 / peridomestic | Trypanosomatidae sp. | 100% | 99.8% |
| PX260238 | *Sciopemyia sordellii* | Jan-25 / peridomestic | *Trypanosoma* sp. | 100% | 100% |

matidae were represented (Fig. 5). Amplification of the mitochondrial *COI* gene was successful for all samples from both sylvatic and peridomestic settings, confirming DNA integrity and serving as an endogenous control [Supplementary data (Fig. 3)].

## DISCUSSION

Sand flies are widely recognised as vectors of *Leishmania*, but our results demonstrate that they can also harbour a broader spectrum of trypanosomatids, including poorly explored lineages. In the MT, we detected both *L. infantum* and *L. braziliensis*, as well as non-*Leishmania* taxa such as *H. samuelpessoai*, *N. esmeraldas*, a representative of Strigomonadinae, and *Trypanosoma* sp. from the Anura clade. In peridomestic settings, we additionally identified a lineage related to the genus *Sergeia* and another *Trypanosoma* sp. from the Anura clade. We also recorded *Sc. birali*, representing the first occurrence of this recently described species in Minas Gerais. [34] Collectively, these findings reveal complex parasite-insect associations across sylvatic and peridomestic environments and reinforce the need to study sand flies beyond the narrow context of *Leishmania* ecology.

The species accumulation curve in the MT reached asymptote, and the observed richness (21 species) closely matched the *Chao*1 estimator, indicating that the sampling effort was sufficient to characterise the local community. Comparable surveys in nearby municipalities, such as Baldim,[35] and Jaboticatubas,[36] reported similar faunistic patterns, reinforcing that the richness observed in our study falls within the expected range for this region of Minas Gerais.

Species of *Pintomyia* (*Pintomyia*), particularly *Pi. fischeri* and *Pi. pessoai*, have been implicated as vectors of *L. infantum* and *L. braziliensis*, respectively,[18,37,38,39,40] and are known to feed on humans and domestic animals.

[41,42] In MT, *Pi. pessoai* and *Pi. christenseni* were found carrying *L. infantum* DNA, suggesting their possible participation in wild-synanthropic transmission cycles of this parasite. Conversely, the role of species within *Pintomyia* (*Pifanomyia*) in *Leishmania* transmission remains debated. *Pintomyia monticola*, probably the most plausible putative vector of this subgenus, has been reported carrying DNA of both *L. infantum* and *L. braziliensis* in southeastern Brazil.[43,44] In MT, *Pi. monticola* was positive for *L. braziliensis* DNA, and notably, it was observed feeding on humans during the day (9-11 a.m.). Such anthropophilic behaviour has been described previously.[45,46,47] Vector competence studies are required to confirm whether this species can sustain late-stage infections of *Leishmania*, thus validating its vectorial role.

Another species of interest, *Pa. barretoi*, also tested positive for *L. infantum*. Ecological information on this species is scarce, as is the case for other members of *Psathyromyia* (*Forattiniella*). Among them, only *Pa. aragaoi* has been reported feeding on humans[48,49] and testing PCR-positive for *L. braziliensis*.[50,51] Our findings suggest that *Pa. barretoi* may have fed on hosts susceptible to *L. infantum* infection and could participate in wild or synanthropic transmission cycles of this parasite in MT. This reinforces that non-traditional sand fly species can also harbour *Leishmania* DNA, although the epidemiological significance of such detections remains uncertain.

Strikingly, the identification of non-*Leishmania* trypanosomatids broadens the spectrum of parasite diversity associated with sand flies. Reports of *Herpetomonas* and *Crithidia* in Brazilian sand flies have increased over the past decade,[8,9] while the occurrence of *Novymonas* is, to our knowledge, described here for the first time in association with a sand fly. Although *Herpetomonas* is generally considered of limited epidemiological relevance,

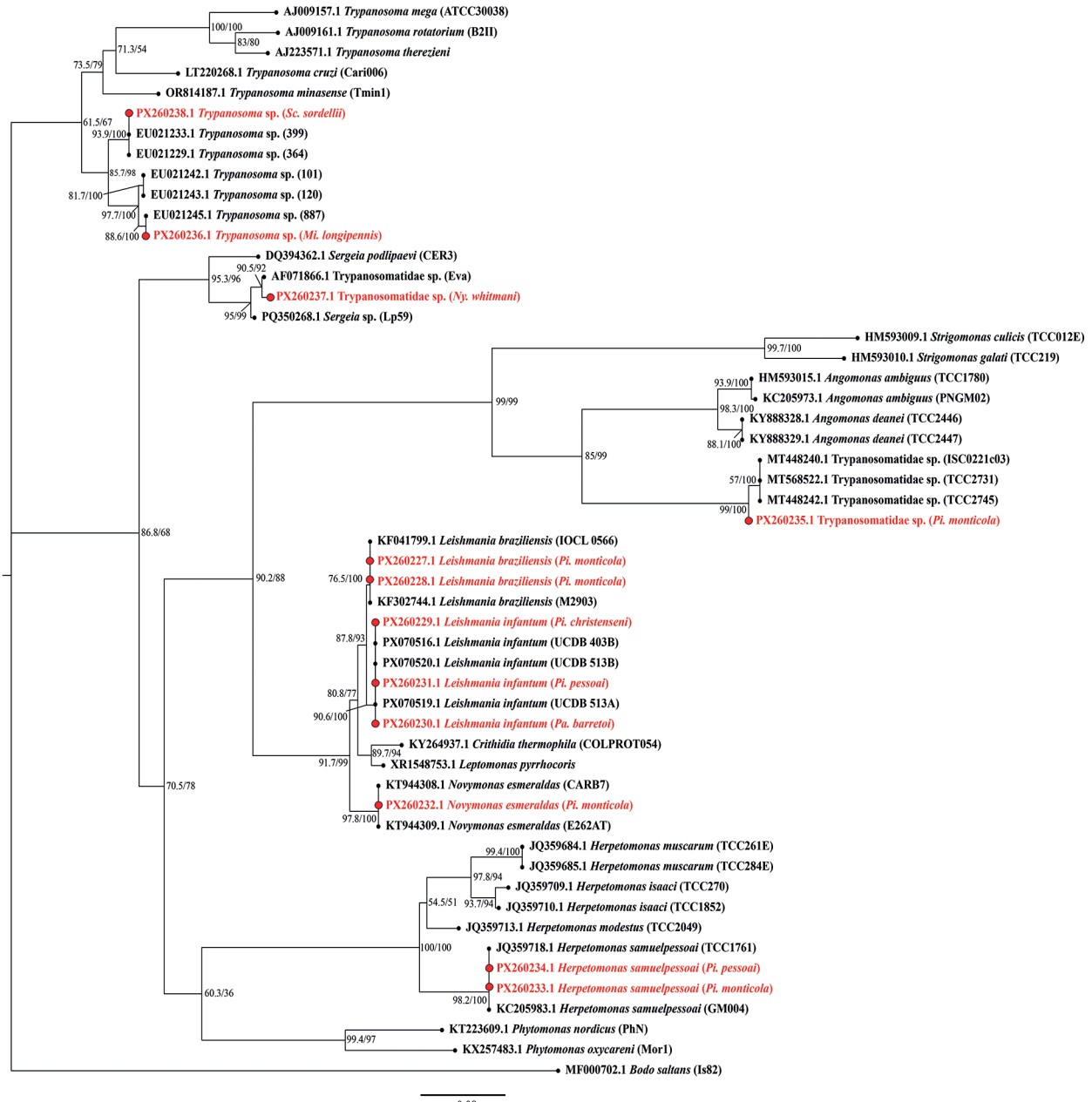

Fig. 3: maximum likelihood (ML) phylogenetic tree of trypanosomatids based on the *18S rRNA* gene (V7-V8 region). Node support values are shown as SH-aLRT (%) / ultrafast bootstrap (UFBoot) replicates (%), based on 1,000 iterations. The tree was rooted with *Bodo saltans* (MF000702.1). Sequences generated in this study are highlighted in red.

it has been reported infecting Egyptian rats[52] and even in immunodepressed humans,[53] fuelling the debate on the true extent of their monoxenous origin. In contrast, *N. esmeraldas*, the only species currently recognised in the genus, was originally described in the hemipteran *Niesthrea vincentii* (Hemiptera: Rhopalidae) from Ecuador.[54] Phylogenetic and molecular evidence place *Novymonas* within the subfamily Leishmaniinae,[55] providing valuable insights into the evolutionary origins of dixenous parasites within this lineage.[56] In our study, BLAST analysis and both ML and Bayesian phylogenetic inferences consistently positioned the parasite detected

in *Pi. monticola* as *N. esmeraldas*, supporting this taxonomic assignment. Nonetheless, given the substantial geographic distance between the type locality in Ecuador and our study area in Brazil, further molecular analyses using additional markers such as the full length of *18S rDNA*, *GAPDH* and *SL-RNA* genes would be necessary to confirm species identity. Such analyses could not be performed here due to limited DNA availability.

*Pintomyia monticola* was also found positive for a trypanosomatid identified as belonging to the subfamily Strigomonadinae.[57,58] This group currently comprises the genera *Angomonas*, *Kentomonas*, and *Strigomonas*.

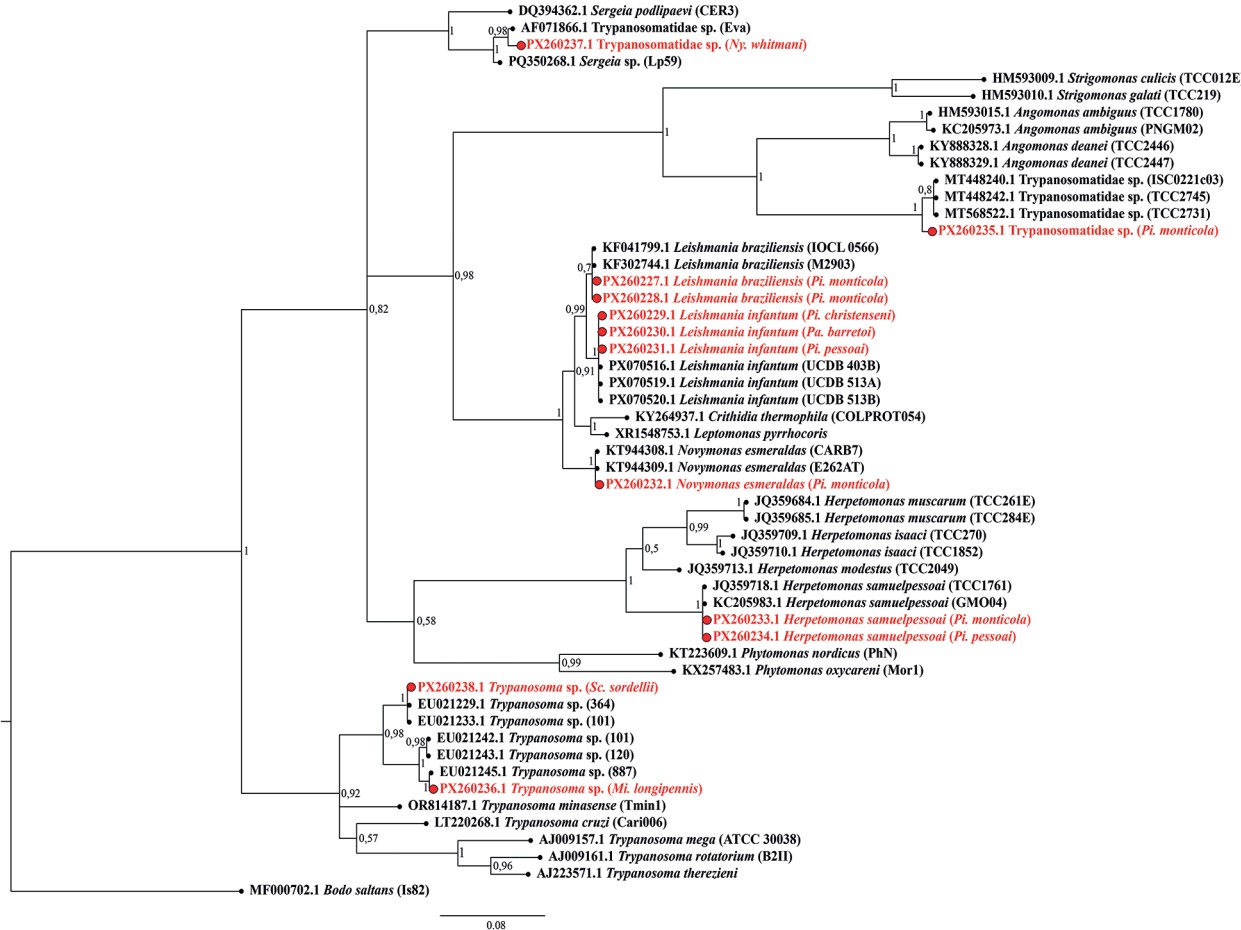

Fig. 4: Bayesian inference (BI) phylogenetic tree of trypanosomatids based on the *18S rRNA* gene (V7-V8 region). Two independent runs of four chains each were performed for 5,000,000 generations, sampling every 100 generations and discarding the first 25% as burn-in. Posterior probabilities are indicated at nodes. The tree was rooted with *Bodo saltans* (MF000702.1). Sequences generated in this study are highlighted in red.

Parasites of this lineage have been described in the Malpighian tubules, haemolymph, hemocoel, and particularly in the midgut of dipterans, which is considered the preferential site for multiplication and colonisation.[59] *Strigomonas culicis*, for example, can colonise the insect midgut, subsequently invade the hemocoel, and eventually reach the salivary glands.[59,60] In sand flies, a representative of *S. galati* has previously been reported in *Lu. almerioi* from Mato Grosso do Sul, Brazil.[57] Interestingly, the parasite identified in *Pi. monticola* in our study was positioned in both ML and Bayesian inferences among sequences labelled as "Trypanosomatidae sp." obtained from *Musca domestica* (Diptera: Muscidae) in Brazil (GenBank: MT448242.1 and MT568522.1) and from an unidentified dipteran in Barbados (MT448240.1). This cluster appears to be phylogenetically close to *Angomonas*, but due to the uncertain taxonomic resolution, we reported our detection as "Trypanosomatidae sp" like those sequences previously deposited.

A parasite closely related to the genus *Sergeia* (here referred to as "Trypanosomatidae sp"; GenBank PX260237) was detected in a female of *Ny. whitmani* from a peridomestic site. Species of *Sergeia*, such as *S. podlipaevi*, have been reported in the gut and Malpighi-an tubules of two biting midge species, *Culicoides (Oecacta) festivipennis* and *C. (Oecacta) truncorum*. Experimental infections in *C. (Monoculicoides) nubeculosus* demonstrated that by the fifth day post-infection, flagellates complete their development in the gut and most localise to the Malpighian tubules.[61] However, *Sergeia* parasites are probably not restricted to biting midges, as phylogenetic analyses revealed an affinity between *S. podlipaevi* and Trypanosomatidae sp "strain EVA", isolated in Venezuela from *Lu. evansi*.[61] Altogether, these results indicate that *Sergeia* parasites may not be restricted to biting midges but instead occupy a broader ecological niche that includes sand flies, raising questions about the evolutionary pathways and host associations of these trypanosomatids.

The detection of *Mi. longipennis* and *Sc. sordellii* carrying *Trypanosoma* sp DNA, previously described in anurans,[62] corroborates the feeding habits reported for species of these genera, which are frequently associated with cold-blooded hosts, despite occasional records of feeding on warm-blooded animals.[63,64] Several studies have documented the occurrence of *Trypanosoma*[65,66,67] and *Leishmania*[51,68,69] in *Micropygomyia* species, suggesting that they are susceptible to trypanosomatid in-

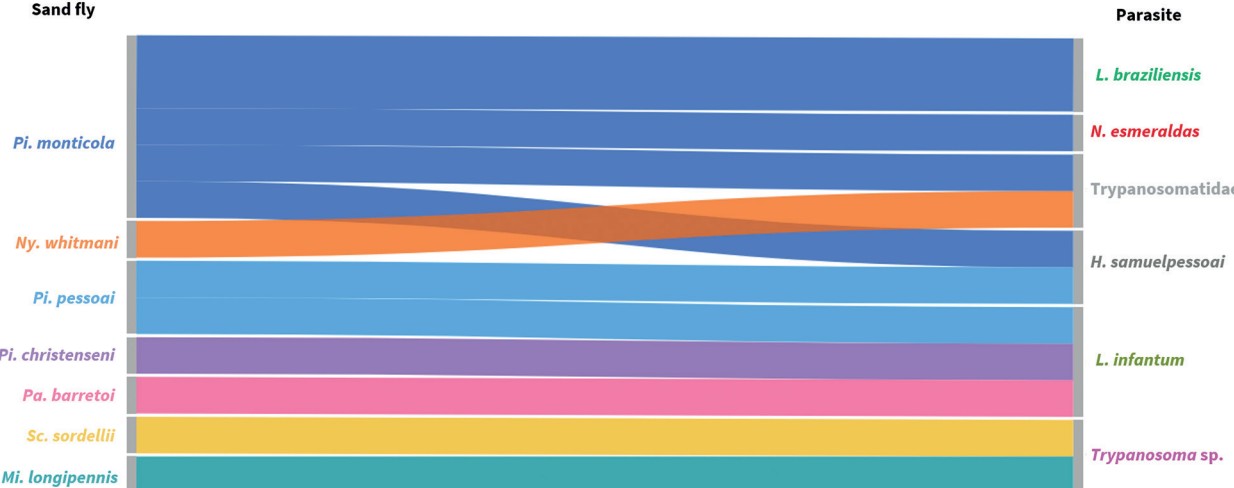

Fig. 5: bipartite interaction diagram (Sankey diagram) between sand fly species and Trypanosomatidae identified during the study in the Serra do Cipó, Minas Gerais, Brazil.

fections. For *Trypanosoma*, the evidence clearly indicates that both *Micropygomyia* and *Sciopemyia* can host these parasites. In contrast, for *Leishmania*, the evidence remains insufficient to establish a vectorial role, which is still a matter of debate.

From an ecological perspective, the Serra do Cipó district, where MT is located, represents a landscape in which sylvatic boundaries are blurred by human settlement, forest fragmentation, and the overlap of domestic and wild reservoirs. Such contexts not only sustain the persistence of *Leishmania* transmission but may also facilitate incidental associations with other trypanosomatids. Species such as *Lu. longipalpis*, *Ny. whitmani*, and *Ev. cortelezzii* are well known for their ability to exploit anthropised habitats, and their presence in peridomestic areas reinforces their role in maintaining leishmaniasis cycles.[5,16,70] The detection of non-*Leishmania* trypanosomatids further suggests that sand flies may act as ecological "hubs", transiently interacting with parasites whose life cycles remain poorly understood. Whether these associations represent accidental infections, dead-end interactions, or potential adaptive processes remains an open question. In conclusion, our findings demonstrate that although the medical relevance of most of these parasites remains uncertain, the hidden diversity of trypanosomatids highlights the need to look beyond *Leishmania* in sand fly research, adopting a broader ecological perspective.

## ACKNOWLEDGEMENTS

To the Fiocruz Network of Technological Platforms at Instituto René Rachou (Fiocruz Minas) for DNA sequencing services.

## AUTHORS' CONTRIBUTION

FDR acquired funding, revised the manuscript and was the primary contributor to the conceptualisation and design of the study; API and FDR wrote the original version of the manuscript, prepared figures and tables, analysed and interpreted data; API and ACCM performed experiments; FDR, GMLC, MLF, DMA, LGD and JDAF conducted field collections; FDR and JDAF supported the methodologies and supervised the study. All authors approved the final version of the manuscript. The funders did not participate in the study's design, data collection and analysis, publishing decisions, or manuscript preparation. None of the authors has a conflict of interest to disclose.

## DATA AVAILABILITY

All relevant data supporting the findings of this study are included within the manuscript and its Supplementary data. DNA sequences generated in this study have been deposited in GenBank under accession numbers PX260227 - PX260238. Additional datasets generated and/or analysed during the current study are available from the corresponding author upon reasonable request.

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

# OPEN PEER REVIEW

Memórias do IOC thanks the anonymous reviewers for their contribution to the peer review of this work.

## FIRST REVIEW ROUND

### REVIEWERS' COMMENTS

### REVIEWER #1

Dear Authors,

The manuscript "Beyond Leishmania: Hidden trypanosomatid diversity reveals complex parasite-sand fly networks in Southeastern Brazil" (MIOC-2025-0260) fits very well within the scope of Memórias do Instituto Oswaldo Cruz. After a detailed evaluation, my major suggestions are as follows:

1. Include a discussion about the absence of a blood source analysis in engorged sand flies;

2. Reduce the number of tables in the main text to two or three, condensing the results (see examples in: https://doi.org/10.1590/0074-02760200157); and

3. Give more attention to Crithidia, the most frequently detected genus in trypanosomatid surveys and recently associated with human disease (https://doi.org/10.3201/eid2511.181548).

However, it is understandable that there is not enough time to a blood source evaluation in the submitted manuscript. Some small points must be solved before the final acceptance:

1. Keywords: Keywords should offer alternative ways for readers to find studies on similar topics. Please consider adding keywords that do not overlap with those in the title.

2. Abstract (line 27): I believe "genera" fits better than "subfamilies", but this is just a suggestion.

3. References: In-text citations should be represented by numbers corresponding to the reference list, which must be arranged in the order of citation. Please, adjust to the journal's formatting guidelines.

4. Lines 50–51: The mention "in targeted surveillance of Leishmania" reads more naturally than "Leishmania-targeted."

5. Line 83: Is a single Shannon trap collection in July sufficient for this type of survey? Since different sand fly species were captured in the two traps, additional collections in other months could better represent sand fly diversity, don't you think?

6. Line 89: Similarly, the peridomestic survey may underestimate sand fly diversity.

7. Natural Infection Analysis: This could be performed using more sensitive methods, such as inoculation of gut fragments into NNN/Schneider's medium or other media more efficient for isolating monoxenous trypanosomatids and Trypanosoma spp. Have you considered this approach?

8. Line 106: Considering that this is a general survey for trypanosomatids, would molecular detection in males not also be of interest? Monoxenous trypanosomatids are believed to be transmitted among insects mainly through water, oviposition, predation or coprophagy, and thus are not exclusively associated with blood-feeding specimens. However, screening male sand flies would not provide evidence for vertebrate-associated transmission.

9. Lines 115 and 121: The PCR reactions for COI and 18S rRNA are described very briefly. Please provide additional methodological details, including cycling conditions, temperatures, and reaction steps.

10. Line 245: A reference to "personal communication" may be more appropriate when discussing the presence of L. infantum in mammals from the same study area.

11. Line 278: Mentions of Figures 3 and 4 in the Discussion section are unnecessary.

12. References (line 347): They should be numbered and listed in order of appearance in the text.

13. Figure 1: The study area could be more clearly illustrated with a map, similar to the illustration used in your group's 2024 publication (https://doi.org/10.1371/journal.pone.0302567) and other studies cited in the manuscript.

### REVIEWER #2

The study addresses the infection of sand flies by Leishmania among other trypanosomatids, providing new findings on the circulation of these parasites in phlebotomine, including new records of sand fly species in Minas Gerais.

Abstract
Line 15-17: I suggest that the authors better describe the methods used.
Introduction
Line 53-57: Citations are required in these sentences.
Methods
Why did the authors not also submit the dissected females for PCR?

Were any engorged females observed?

Results

Tables: Table 1 already represents the fauna results well, I would suggest that Table 2 could be supplementary, as well as Table 3 and Table 5.

Table 4: I suggest that authors include the percentage of identity and query cover found in genbank.

In the tables, the authors report the finding of Sc. birali, a recently described species. I suggest commenting on this new record for the state of Minas Gerais in the results and discussion.

Figure 2. I suggest that the authors revise Figure 2. A better analysis would be across environments, rather than collection months. For example, an analysis of the richness between the forest and peridomestic environments.

Regarding phylogenetic analysis, wouldn't it be interesting to include Porcisia sequences?

Discussion

Regarding the discussion section, I am satisfied with the text presented, however, the authors could also explore and discuss a little more the results of the species richness found, as well as the new finding of Sciopemyia birali in Minas Gerais.

References

The references are adequate

**REVIEWER #3**

General Comments

Abstract was presented properly with a summary of context, methods, results and main conclusions of the study.

The manuscript has original results obtained with consistent and well-established sampling and molecular techniques but bring an alternative view for neglected Trypanosomatidae in sand fly surveys. The findings on this purpose bring the relevance of the manuscript.

Methodology is appropriated but suggestion on its presentation were made for enhancing the manuscript.

Results and main findings are well discussed.

Figures 2-4 may have higher resolution. Suggestions on figure 5 were made in specific comments.

Specific Comments

Comments on methodology, results and discussion

Line 16: remove "(MT)"

Lines 49-51: It is not clear whether a broader range of parasite vectors interactions hampers the interpretation on Leishmania targeted surveillance. Please review the statement.

Line 52: Reference needed.

Lines 57-59: Review the sentence considering that, from an evolutionary standpoint, it is unlikely that a sand fly species would develop a novel ability or environmental adaptation, but rather that such traits would be inherent outcomes of its evolutionary history.

Line 66: Consider the substitution of "to characterize" by "to explore" once the study design description is not clear in terms of time scale an area coverage, even though ecological analyses adequately explore sampling efforts.

Line 70: Please, consider including the souce of the information or excluding it from the text.

Lines 83-86: in line 85 - "were separated using Castro aspirator" - it could seem to be out of sense considering collections in Shannon traps. Please review the text considering both sentences.

Lines 86-88: For me, it was not clear whether dissected sand flies were included or not in the molecular experiments. Please review the sentence.

Lines 73-93:

Sampling efforts presented seem to have some limitations in terms of time scale regarding the gap between MT and peridomestic survey hampering comparisons among the two collection series. It is also not clear whether there is a rational of geographic (special or ecological) distribution of collection sites. Do the collection sites represent the Serra do Espinhaço or even the Mata da Tapera?

It would be nice to see this in the text or maybe in a map presenting the characteristics of the area and the collection sites.

Line 96: what is the Leishmaniasis Group?

Line 106: Review the sentence: "Females not used for direct dissection were processed individually".

Line 112: Review "and extraction blanks".

Line 117: Consider replacing "under" by "and".

Line 118: Review the description of the NTC considering that it should be a tube with the same mix but, instead of a sample, a volume of negative solution should be added.

Lines 111-119: Protocols presented do not guarantee the integrity of pathogens DNA. In line 114, consider including "integrity of sand fly DNA".

Line 129: Remove citation "Rêgo et al. 2015".

Lines 129-131: what were the parameters for consensus preparation? (e.g. phred quality?)

Were there any ambiguous sites? If so, how were they treated?

Were the primer sequences removed?

Line 134: There is a miss concept in "were combined with homologous sequences".

Sstructure suggestion: "Sequences for phylogenetic analyses were mined from Genbank by Blast filtering for Leishmaniinae, Phytomonadinae, Strigomonadinae, and Trypanosomatinae, with Bodo saltans included as the outgroup"

Line 138: Why V7V8 low quality sequences were not removed in previous steps (line 129)?

Line 139: What is the electropherogram inspection and how was it made? Consider explaining the steps.

Full alignment could be replaced by "global alignment".

Line 176: Review abbreviation of scientific names allover the text as well as species authors.

Line 186: Review grammar in sentence structure and the use of "as".

Lines 157-160: Parameters for Sankey diagram are not clear. Looking the graphic in page 35, it is clearly possible to get more direct e simpler relationships. Review the graphic.

Line 277: Remove "Sanger sequencing". Do the authors intend to say "Blast analyses"? This could be a suggestion.

Line 281: Please, include "e.g." within the parentheses or explain why only those three targets were proposed as a solution.

## AUTHORS' RESPONSE TO THE REVIEWERS

December 3, 2025

Manuscript ID: MIOC-2025-0260

Beyond Leishmania: Hidden trypanosomatid diversity reveals complex parasite-sand fly networks in Southeastern Brazil

Dear Dr. Cupolillo,

We sincerely thank you and the reviewers for the thorough and constructive evaluation of our manuscript. We carefully considered all comments and incorporated the suggested revisions to improve clarity, methodological transparency, and the overall scientific quality of the work. In several instances, the reviewers raised important conceptual and methodological points that helped us refine the presentation of our results and strengthen key interpretations.

Below, we provide a detailed, point-by-point response to all comments. Reviewer remarks are presented in italics, followed by our responses. For clarity, we refer to page and line numbers in the track change version of the manuscript. Whenever we slightly disagreed with a suggestion, we explain our reasoning while still ensuring that the final version addresses the reviewer's concern.

We hope that the revised manuscript meets the expectations of the editorial board and reviewers, and we remain at your disposal for any additional adjustments needed.

Sincerely,

Felipe Dutra Rêgo

Instituto René Rachou - Oswaldo Cruz Foundation

Reviewer comments to the authors

Reviewer #1:

General comment:

Dear Authors, the manuscript "Beyond Leishmania: Hidden trypanosomatid diversity reveals complex parasite-sand fly networks in Southeastern Brazil" (MIOC-2025-0260) fits very well within the scope of Memórias do Instituto Oswaldo Cruz.

Specific comments:

1. Include a discussion about the absence of a blood source analysis in engorged sand flies; However, it is understandable that there is not enough time to a blood source evaluation in the submitted manuscript.

Authors' comment: Thank you for your observation. No engorged females were collected during the sampling campaigns, which made blood-meal source analyses unfeasible in this study. We have now clarified this point in the Result section (lines 191-192) to avoid any impression of a methodological gap. Given the absence of engorged specimens, we believe that a more detailed discussion on this topic would fall outside the scope of the present work.

2. Reduce the number of tables in the main text to two or three, condensing the results (see examples in: https://doi.org/10.1590/0074-02760200157);

Authors' comment: In accordance with your recommendation, we have reduced the number of tables presented in the main text. The tables containing the monthly distribution of sand fly species across the sampling campaigns and in peridomestic settings (former Table 2 and 5) have been moved to the Supplementary Material, as it provides detailed temporal information that is not essential for the interpretation of the main ecological and molecular findings.

3. Give more attention to Crithidia, the most frequently detected genus in trypanosomatid surveys and recently associated with human disease (https://doi.org/10.3201/eid2511.181548).

Authors' comment: Thank you for this suggestion. However, no sequences obtained in our study clustered with Crithidia spp. in either BLAST searches or phylogenetic analyses. For this reason, a dedicated discussion on Crithidia would be speculative and not directly supported by our results. We therefore limited the discussion to the parasite groups detected in the study area, ensuring accuracy and coherence with the molecular identifications.

Some small points must be solved before the final acceptance:

1. Keywords: Keywords should offer alternative ways for readers to find studies on similar topics. Please consider adding keywords that do not overlap with those in the title.

Authors' comment: We have revised the keywords to include broader and more searchable terms, improving discoverability and alignment with commonly indexed descriptors. The updated keywords are: Phlebotominae; Leishmania; Trypanosomatidae; Phylogenetic analysis; Vector-parasite interactions.

2. Abstract (line 27): I believe "genera" fits better than "subfamilies", but this is just a suggestion.

Authors' comment: The sentence has been updated accordingly (line 28).

3. References: In-text citations should be represented by numbers corresponding to the reference list, which must be arranged in the order of citation. Please, adjust to the journal's formatting guidelines.

Authors' comment: We have reformatted all in-text citations to the numerical style required by the journal and reorganized the reference list according to the order of appearance in the manuscript. The reference section now fully complies with the formatting guidelines of Memórias do Instituto Oswaldo Cruz.

4. Lines 50–51: The mention "in targeted surveillance of Leishmania" reads more naturally than "Leishmania-targeted."

Authors' comment: The phrasing has been amended accordingly, and "in targeted surveillance of Leishmania" has replaced "Leishmania-targeted" in the revised manuscript (line 47-50).

5. Line 83: Is a single Shannon trap collection in July sufficient for this type of survey? Since different sand fly species were captured in the two traps, additional collections in other months could better represent sand fly diversity, don't you think?

Authors' comment: We agree that additional Shannon trap collections could potentially capture further species and contribute to a more comprehensive faunal assessment. However, in this study the Shannon trap was used as a complementary method to obtain live females for dissection rather than as a primary tool for estimating species richness.

Importantly, rarefaction and extrapolation analyses indicated that species richness reached asymptote, with observed and estimated richness (Chao1) coinciding and confidence intervals overlapping. This demonstrates that the sampling effort based on CDC traps was (apparently) sufficient to capture local diversity patterns, even with only one Shannon collection.

6. Line 89: Similarly, the peridomestic survey may underestimate sand fly diversity.

Authors' comment: We agree that the peridomestic survey, conducted during a single sampling campaign, does not aim to represent the full diversity of sand flies in peri-urban environments of the Serra do Cipó region. The core objective of the study was the multi-campaign ecological and parasitological assessment of the Mata da Tapera forest fragment, and the peridomestic sampling served as a complementary component to explore parasite-vector interactions at the forest-household interface.

The reduced sampling effort may underestimate local species richness; however, this limitation does not affect the main conclusions of the study, which rely on the more comprehensive dataset from the forest fragment. We have clarified this point in the revised manuscript (line 227-228). Importantly, expanded peridomestic surveys are currently underway in the region, aiming to generate a more complete inventory of sand fly fauna and to refine the preliminary patterns presented here.

7. Natural Infection Analysis: This could be performed using more sensitive methods, such as inoculation of gut fragments into NNN/Schneider's medium or other media more efficient for isolating monoxenous trypanosomatids and Trypanosoma spp. Have you considered this approach?

Authors' comment: We agree that isolation of gut fragments in culture media such as NNN, LIT or Schneider's medium can be useful for recovering certain trypanosomatids. However, in sand flies this approach often faces substantial limitations. The midgut frequently contains high bacterial loads, and despite the use of antibiotics, contamination frequently overgrows cultures before trypanosomatids can be detected. In addition, monoxenous lineages and some Trypanosoma spp. differ markedly in their ability to grow in vitro, leading to low and heterogeneous recovery rates.

Because of these constraints, there is currently no standardized protocol for natural infection analysis in phlebotomine sand flies. Studies variably rely on midgut dissection alone, dissection combined with PCR, direct PCR of whole females, or attempted culture isolation, each with specific drawbacks.

8. Line 106: Considering that this is a general survey for trypanosomatids, would molecular detection in males not also be of interest? Monoxenous trypanosomatids are believed to be transmitted among insects mainly through water, oviposition, predation or coprophagy, and thus are not exclusively associated with blood-feeding specimens. However, screening male sand flies would not provide evidence for vertebrate-associated transmission.

Authors' comment: Thank you for this interesting comment. We agree that screening male sand flies could provide valuable insights into the ecology of monoxenous trypanosomatids, a topic that remains largely unexplored in the literature. We recognize the potential of this approach and consider it a promising avenue for future investigations.

9. Lines 115 and 121: The PCR reactions for COI and 18S rRNA are described very briefly. Please provide additional methodological details, including cycling conditions, temperatures, and reaction steps.

Authors' comment: The COI (lines 124-129) and 18S rRNA (lines 134-139) PCRs were performed following the original protocols cited in the manuscript. To improve clarity, we have now added the essential cycling parameters (initial denaturation, annealing and extension temperatures, number of cycles, and reaction volumes) while keeping the description concise to avoid unnecessary duplication of previously established methods.

10. Line 245: A reference to "personal communication" may be more appropriate when discussing the presence of L. infantum in mammals from the same study area.

Authors' comment: The sentence has been revised to refer to this information as a personal communication in the updated manuscript (line 270).

11. Line 278: Mentions of Figures 3 and 4 in the Discussion section are unnecessary.

Authors' comment: The unnecessary mentions of Figures 3 and 4 in the Discussion have been removed in the revised version of the manuscript.

12. References (line 347): They should be numbered and listed in order of appearance in the text.

Authors' comment: As noted above, all in-text citations have been converted to numerical format and the reference list has been reordered to follow the sequence of appearance in the manuscript.

13. Figure 1: The study area could be more clearly illustrated with a map, similar to the illustration used in your group's 2024 publication (https://doi.org/10.1371/journal.pone.0302567) and other studies cited in the manuscript.

Authors' comment: Following your recommendation, Figure 1 has been updated to include a clearer map of the study area. The revised figure provides improved geographic context and visualization of the sampling sites.

Reviewer #2:
General comment:
The study addresses the infection of sand flies by Leishmania among other trypanosomatids, providing new findings on the circulation of these parasites in phlebotomine, including new records of sand fly species in Minas Gerais.

Specific comments:
Abstract
Line 15-17: I suggest that the authors better describe the methods used.
Authors' comment: We have revised the sentence to provide a clearer description of the methodological approach in the abstract. The updated version (line 15-18) now specifies that sampling was followed by midgut dissection and molecular screening for trypanosomatids.

Introduction
Line 53-57: Citations are required in these sentences.
Authors' comment: We have added appropriate citations to support these statements in the Introduction. The revised version now includes references discussing the influence of periurban environments and habitat fragmentation on sand fly ecology and on the circulation of both Leishmania and non-Leishmania trypanosomatids (line 51-55).

Methods
Why did the authors not also submit the dissected females for PCR?
Were any engorged females observed?
Authors' comment: The females selected for midgut dissection (n = 105) were examined exclusively for the presence of flagellates, while molecular screening was conducted on a separate set of ethanol-preserved females.

Midgut dissection typically leaves very limited biological material for reliable DNA extraction, which often results in low yield and reduced amplification success. Although PCR-based methods are generally sensitive, there is no consistent evidence that performing PCR on dissected midguts improves detection rates compared to screening intact females preserved specifically for molecular analyses. For this reason, PCR was applied only to intact ethanol-preserved females, ensuring higher DNA integrity and minimizing sample loss.

No engorged females were observed in any of the collections, which made blood-meal analysis or PCR-based identification of vertebrate hosts unfeasible. This point has been clarified in the revised manuscript (line 191-192).

Results

Tables: Table 1 already represents the fauna results well, I would suggest that Table 2 could be supplementary, as well as Table 3 and Table 5.

Authors' comment: Thank you for this observation. Following your suggestion, the table containing the monthly distribution of sand flies (former Table 2), the table summarizing dissected females, and the table listing peridomestic species (former Table 5) have been moved to the Supplementary Materials 2 and 4, respectively.

We retained only three tables in the main text, those summarizing species composition, dissected females, and molecular identifications of trypanosomatid-positive specimens, because they contain essential information for interpreting the main ecological and parasitological findings.

Table 4: I suggest that authors include the percentage of identity and query cover found in genbank.

In the tables, the authors report the finding of Sc. birali, a recently described species. I suggest commenting on this new record for the state of Minas Gerais in the results and discussion.

Authors' comment: We have now added the percentage identity and query coverage values obtained from GenBank to Table 3 to improve transparency in the molecular identifications. We also included a brief comment in the Discussion section highlighting the record of Sciopemyia birali, which represents a new occurrence for the state of Minas Gerais.

Figure 2. I suggest that the authors revise Figure 2. A better analysis would be across environments, rather than collection months. For example, an analysis of the richness between the forest and peridomestic environments.

Authors' comment: We agree that comparing species richness between forest and peridomestic environments can be informative; however, such an analysis requires equivalent sampling effort across sites. In our study, five sampling campaigns were conducted in the Mata da Tapera, whereas only a single campaign was carried out in peridomestic areas. Because richness estimates obtained through rarefaction and extrapolation are highly sensitive to sampling effort, a direct comparison between the two environments would not be methodologically appropriate.

For this reason, Figure 2 focuses exclusively on the Mata da Tapera dataset, where the multi-campaign structure allows proper accumulation, interpolation, and extrapolation analyses. This approach enables us to evaluate sampling completeness and to verify that species richness reached asymptote, supporting the robustness of the ecological inferences drawn from that site.

Regarding phylogenetic analysis, wouldn't it be interesting to include Porcisia sequences?

Authors' comment: We agree that Porcisia, and other groups like Endotrypanum and Leishmania (Mundinia) represent relevant lineages within Leishmaniinae. However, none of the sequences obtained in our study clustered near these groups in preliminary BLAST analyses, and the phylogenetic placement of our isolates was consistently resolved within other well-supported clades. Including additional reference sequences from these genera would therefore not modify the inferred relationships or improve the interpretation of our results, while substantially increasing the size and complexity of the tree. For these reasons, we retained a focused phylogenetic framework containing the taxa necessary to robustly position the sequences generated in this study.

Discussion

Regarding the discussion section, I am satisfied with the text presented, however, the authors could also explore and discuss a little more the results of the species richness found, as well as the new finding of Sciopemyia birali in Minas Gerais.

Authors' comment: We expanded the Discussion by adding a concise paragraph that places our species richness findings in the context of comparable surveys in Minas Gerais (e.g., Baldim and Jaboticatubas) and highlights the adequacy of our sampling effort in the Mata da Tapera forest fragment (lines 258-263).

Reviewer #3:

General comment:

Abstract was presented properly with a summary of context, methods, results and main conclusions of the study.

The manuscript has original results obtained with consistent and well-established sampling and molecular techniques but bring an alternative view for neglected Trypanosomatidae in sand fly surveys. The findings on this purpose bring the relevance of the manuscript.

Methodology is appropriated but suggestion on its presentation were made for enhancing the manuscript.
Results and main findings are well discussed.
Figures 2-4 may have higher resolution. Suggestions on figure 5 were made in specific comments.

Specific comments:
Line 16: remove "(MT)"
Authors' comment: The abbreviation "(MT)" has been removed from the Abstract as suggested.
Lines 49-51: It is not clear whether a broader range of parasite vectors interactions hampers the interpretation on Leishmania targeted surveillance. Please review the statement.
Authors' comment: We have revised the sentence to clarify that the presence of non-Leishmania trypanosomatids does not hinder surveillance itself, but rather increases the complexity of interpreting molecular results when broad-range markers are used (lines 47-50).

Line 52: Reference needed.
Authors' comment: A reference has been added to support this statement in the revised manuscript (lines 51-55).

Lines 57-59: Review the sentence considering that, from an evolutionary standpoint, it is unlikely that a sand fly species would develop a novel ability or environmental adaptation, but rather that such traits would be inherent outcomes of its evolutionary history.
Authors' comment: Thank you for the observation. The sentence has been revised to avoid implying recent evolutionary changes in sand fly species. The updated version clarifies that these species are naturally suited to anthropized environments, and that the uncertainty concerns their role in the ecology of non-Leishmania trypanosomatids (line 55-61).

Line 66: Consider the substitution of "to characterize" by "to explore" once the study design description is not clear in terms of time scale an area coverage, even though ecological analyses adequately explore sampling efforts.
Authors' comment: The term "to characterize" has been replaced with "to explore" in the revised manuscript (line 63).

Line 70: Please, consider including the source of the information or excluding it from the text.
Authors' comment: The information regarding autochthonous cases of canine visceral leishmaniasis has been revised and is now cited as personal communication from the Municipal Health Department (lines 66-67).

Lines 83-86: in line 85 - "were separated using Castro aspirator" - it could seem to be out of sense considering collections in Shannon traps. Please review the text considering both sentences.
Authors' comment: Thank you for noting this ambiguity. The sentence has been revised to clarify that sand flies from CDC traps were removed from the trap cages using a Castro aspirator, whereas those collected in the Shannon trap were aspirated directly from the capture cloth. The revised text now provides a clearer description of both procedures (line 89-91).

Lines 86-88: For me, it was not clear whether dissected sand flies were included or not in the molecular experiments. Please review the sentence.
Authors' comment: We have revised the paragraph to clarify how sand flies were collected from CDC and Shannon traps and to explicitly state that dissected females were not included in the molecular analyses. The revised text now makes the sampling and specimen allocation procedures fully clear (line 91-94).

Lines 73-93: Sampling efforts presented seem to have some limitations in terms of time scale regarding the gap between MT and peridomestic survey hampering comparisons among the two collection series. It is also not clear whether there is a rational of geographic (special or ecological) distribution of collection sites. Do the collection sites represent the Serra do Espinhaço or even the Mata da Tapera? It would be nice to see this in the text or maybe in a map presenting the characteristics of the area and the collection sites.
Authors' comment: We have clarified the rationale behind the sampling design by specifying that the Mata da Tapera was the primary ecological survey site, sampled across five campaigns, while the peridomestic collections represent a single preliminary survey intended to document species presence at the forest-household interface rather than to enable direct ecological comparisons. We also expanded the description of the geographic context and revised Figure 1 to clearly display the spatial distribution of sampling sites (line 71-88).

Line 96: what is the Leishmaniasis Group?
Authors' comment: The reference to the "Leishmaniasis Group" has been removed, and the sentence now reads simply "Fiocruz Minas," which provides clearer institutional identification (line 102).

Line 106: Review the sentence: "Females not used for direct dissection were processed individually".
Authors' comment: The sentence has been revised to clarify that all females not used for midgut examination were still dissected individually for taxonomic identification prior to molecular screening (Line 111-114).

Line 112: Review "and extraction blanks".
Authors' comment: The sentence has been revised for clarity, and the role of extraction blanks as negative controls is now more clearly stated (line 117-119).

Line 117: Consider replacing "under" by "and".
Authors' comment: The word "under" has been replaced with "and" in the revised manuscript.

Line 118: Review the description of the NTC considering that it should be a tube with the same mix but, instead of a sample, a volume of negative solution should be added.
Authors' comment: The description of the NTC has been revised for clarity. The text now specifies that the no-template control consisted of the complete reaction mix in which sample DNA was replaced by nuclease-free water (line 127-129).

Lines 111-119: Protocols presented do not guarantee the integrity of pathogens DNA. In line 114, consider including "integrity of sand fly DNA".
Authors' comment: The text has been revised to clarify that the COI PCR was used to verify the integrity of sand fly DNA rather than the integrity of pathogen DNA (lines 120-121).

Line 129: Remove citation "Rêgo et al. 2015".
Authors' comment: The citation has been removed.

Lines 129-131: what were the parameters for consensus preparation? (e.g. phred quality?)
Were there any ambiguous sites? If so, how were they treated?
Were the primer sequences removed?
Authors' comment: The description of the sequencing workflow has been expanded to include the criteria used for consensus assembly. (lines 144-149)
Line 134: There is a miss concept in "were combined with homologous sequences".
Sstructure suggestion: "Sequences for phylogenetic analyses were mined from Genbank by Blast filtering for Leishmaniinae, Phytomonadinae, Strigomonadinae, and Trypanosomatinae, with Bodo saltans included as the outgroup"
Authors' comment: The sentence has been revised for clarity (lines 154-157).

Line 138: Why V7V8 low quality sequences were not removed in previous steps (line 129)?
Line 139: What is the electropherogram inspection and how was it made? Consider explaining the steps. Full alignment could be replaced by "global alignment".
Authors' comment: Thank you for the observation. We have revised the text to clarify the sequence-processing workflow. Low-quality regions were removed during the manual inspection of chromatograms prior to consensus generation, and trimAl was subsequently applied only to refine the final alignment. The revised paragraph reflects this sequence of steps (lines 158-165).

Line 176: Review abbreviation of scientific names all over the text as well as species authors.
Authors' comment: The abbreviations used for sand fly genera (e.g., Pi. for Pintomyia) follow the nomenclatural recommendations proposed by Marcondes (2007). A reference to this convention has now been added to the Methods section, and the use of abbreviations and species authors has been checked for consistency throughout the manuscript.

Line 186: Review grammar in sentence structure and the use of "as".
Authors' comment: The sentence has been revised to improve clarity and grammar (line 203-205).

Lines 157-160: Parameters for Sankey diagram are not clear. Looking the graphic in page 35, it is clearly possible to get more direct e simpler relationships. Review the graphic.
Authors' comment: Thank you for the suggestion. We revised the Sankey diagram to improve clarity and reduce line crossings. The figure was regenerated in Flourish using the "Reduce overlaps" sorting option, which provides more direct and visually intuitive vector-parasite links. The Methods section has been updated to describe these parameters (lines 175-177).

Line 277: Remove "Sanger sequencing". Do the authors intend to say "Blast analyses"? This could be a suggestion.
Authors' comment: The term "Sanger sequencing" has been replaced with "BLAST analysis" in the revised manuscript (lines 297-298).

Line 281: Please, include "e.g." within the parentheses or explain why only those three targets were proposed as a solution.

Authors' comment: The sentence has been revised to clarify that the listed markers are examples commonly used in trypanosomatid phylogenetics (line 302).

### SECOND REVIEW ROUND

#### REVIEWERS' COMMENTS

The authors have adequately addressed all points and suggestions raised in the first review.
Congratulations on the manuscript.
One final consideration concerns the citation of a "personal communication" on line 275, which should be avoided or replaced by a reference to another author, if possible.

#### AUTHORS' RESPONSE TO THE REVIEWERS

Dear Dr. Cupolillo,

Thank you very much for your careful evaluation of our revised manuscript entitled "Beyond Leishmania: Hidden trypanosomatid diversity reveals complex parasite-sand fly networks in Southeastern Brazil" (Manuscript ID: MIOC-2025-0260.R1).

We are grateful for the positive feedback from the reviewer and for the final recommendation. As suggested, we have removed the citation of "personal communication" previously indicated on line 275, ensuring full compliance with the journal's editorial standards.

Since no additional technical or methodological adjustments were required, we believe that the manuscript now fully meets all scientific and editorial criteria for publication in Memórias do Instituto Oswaldo Cruz.

We sincerely appreciate the opportunity to revise our work and thank you and the reviewers for the constructive and encouraging evaluation throughout the process.

Kind regards,
Felipe Dutra Rêgo
Instituto René Rachou - Fiocruz Minas

### THIRD REVIEW ROUND

#### REVIEWERS' COMMENTS

No comments.

