## [Reviewer Report · FIRST REVIEW ROUND - REVIEWERS COMMENTS]

## REVIEWER #1

Dear Authors,

The manuscript “Beyond *Leishmania*: Hidden trypanosomatid diversity reveals complex parasite-sand fly networks in Southeastern Brazil” (MIOC-2025-0260) fits very well within the scope of Memórias do Instituto Oswaldo Cruz.

After a detailed evaluation, my major suggestions are as follows:

1. Include a discussion about the absence of a blood source analysis in engorged sand flies;

2. Reduce the number of tables in the main text to two or three, condensing the results (see examples in: https://doi.org/10.1590/0074-02760200157); and

3. Give more attention to *Crithidia*, the most frequently detected genus in trypanosomatid surveys and recently associated with human disease (https://doi.org/10.3201/eid2511.181548).

However, it is understandable that there is not enough time to a blood source evaluation in the submitted manuscript.

Some small points must be solved before the final acceptance:

1. Keywords: Keywords should offer alternative ways for readers to find studies on similar topics. Please consider adding keywords that do not overlap with those in the title.

2. Abstract (line 27): I believe “genera” fits better than “subfamilies”, but this is just a suggestion.

3. References: In-text citations should be represented by numbers corresponding to the reference list, which must be arranged in the order of citation. Please, adjust to the journal’s formatting guidelines.

4. Lines 50–51: The mention “in targeted surveillance of *Leishmania*” reads more naturally than “*Leishmania*-targeted.”

5. Line 83: Is a single Shannon trap collection in July sufficient for this type of survey? Since different sand fly species were captured in the two traps, additional collections in other months could better represent sand fly diversity, don’t you think?

6. Line 89: Similarly, the peridomestic survey may underestimate sand fly diversity.

7. Natural Infection Analysis: This could be performed using more sensitive methods, such as inoculation of gut fragments into NNN/Schneider’s medium or other media more efficient for isolating monoxenous trypanosomatids and *Trypanosoma* spp. Have you considered this approach?

8. Line 106: Considering that this is a general survey for trypanosomatids, would molecular detection in males not also be of interest? Monoxenous trypanosomatids are believed to be transmitted among insects mainly through water, oviposition, predation or coprophagy, and thus are not exclusively associated with blood-feeding specimens. However, screening male sand flies would not provide evidence for vertebrate-associated transmission.

9. Lines 115 and 121: The PCR reactions for COI and 18S rRNA are described very briefly. Please provide additional methodological details, including cycling conditions, temperatures, and reaction steps.

10. Line 245: A reference to “personal communication” may be more appropriate when discussing the presence of *L. infantum* in mammals from the same study area.

11. Line 278: Mentions of Figures 3 and 4 in the Discussion section are unnecessary.

12. References (line 347): They should be numbered and listed in order of appearance in the text.

13. Figure 1: The study area could be more clearly illustrated with a map, similar to the illustration used in your group’s 2024 publication (https://doi.org/10.1371/journal.pone.0302567) and other studies cited in the manuscript.

## REVIEWER #2

The study addresses the infection of sand flies by *Leishmania* among other trypanosomatids, providing new findings on the circulation of these parasites in phlebotomine, including new records of sand fly species in Minas Gerais.

**Abstract**

Line 15-17: I suggest that the authors better describe the methods used.

**Introduction**

Line 53-57: Citations are required in these sentences.

**Methods**

Why did the authors not also submit the dissected females for PCR? Were any engorged females observed?

**Results**

Tables: Table 1 already represents the fauna results well, I would suggest that Table 2 could be supplementary, as well as Table 3 and Table 5.

Table 4: I suggest that authors include the percentage of identity and query cover found in genbank.

In the tables, the authors report the finding of *Sc. birali*, a recently described species. I suggest commenting on this new record for the state of Minas Gerais in the results and discussion.

Figure 2. I suggest that the authors revise Figure 2. A better analysis would be across environments, rather than collection months. For example, an analysis of the richness between the forest and peridomestic environments.

Regarding phylogenetic analysis, wouldn’t it be interesting to include *Porcisia* sequences?

**Discussion**

Regarding the discussion section, I am satisfied with the text presented, however, the authors could also explore and discuss a little more the results of the species richness found, as well as the new finding of *Sciopemyia birali* in Minas Gerais.

**References**

The references are adequate

## REVIEWER #3

**General Comments**

Abstract was presented properly with a summary of context, methods, results and main conclusions of the study.

The manuscript has original results obtained with consistent and well-established sampling and molecular techniques but bring an alternative view for neglected Trypanosomatidae in sand fly surveys. The findings on this purpose bring the relevance of the manuscript.

Methodology is appropriated but suggestion on its presentation were made for enhancing the manuscript.

Results and main findings are well discussed.

Figures 2-4 may have higher resolution. Suggestions on figure 5 were made in specific comments.

**Specific Comments**

Comments on methodology, results and discussion

Line 16: remove “(MT)”

Lines 49-51: It is not clear whether a broader range of parasite vectors interactions hampers the interpretation on *Leishmania* targeted surveillance. Please review the statement.

Line 52: Reference needed.

Lines 57-59: Review the sentence considering that, from an evolutionary standpoint, it is unlikely that a sand fly species would develop a novel ability or environmental adaptation, but rather that such traits would be inherent outcomes of its evolutionary history.

Line 66: Consider the substitution of “to characterize” by “to explore” once the study design description is not clear in terms of time scale an area coverage, even though ecological analyses adequately explore sampling efforts.

Line 70: Please, consider including the souce of the information or excluding it from the text.

Lines 83-86: in line 85 - “were separated using Castro aspirator” - it could seem to be out of sense considering collections in Shannon traps. Please review the text considering both sentences.

Lines 86-88: For me, it was not clear whether dissected sand flies were included or not in the molecular experiments. Please review the sentence.

Lines 73-93: Sampling efforts presented seem to have some limitations in terms of time scale regarding the gap between MT and peridomestic survey hampering comparisons among the two collection series. It is also not clear whether there is a rational of geographic (special or ecological) distribution of collection sites. Do the collection sites represent the Serra do Espinhaço or even the Mata da Tapera? It would be nice to see this in the text or maybe in a map presenting the characteristics of the area and the collection sites.

Line 96: what is the Leishmaniasis Group?

Line 106: Review the sentence: “Females not used for direct dissection were processed individually”.

Line 112: Review “and extraction blanks”.

Line 117: Consider replacing “under” by “and”.

Line 118: Review the description of the NTC considering that it should be a tube with the same mix but, instead of a sample, a volume of negative solution should be added.

Lines 111-119: Protocols presented do not guarantee the integrity of pathogens DNA. In line 114, consider including “integrity of sand fly DNA”.

Line 129: Remove citation “Rêgo et al. 2015”.

Lines 129-131: what were the parameters for consensus preparation? (e.g. phred quality?) Were there any ambiguous sites? If so, how were they treated? Were the primer sequences removed?

Line 134: There is a miss concept in “were combined with homologous sequences”. Sstructure suggestion: “Sequences for phylogenetic analyses were mined from Genbank by Blast filtering for Leishmaniinae, Phytomonadinae, Strigomonadinae, and Trypanosomatinae, with *Bodo saltans* included as the outgroup”

Line 138: Why V7V8 low quality sequences were not removed in previous steps (line 129)?

Line 139: What is the electropherogram inspection and how was it made? Consider explaining the steps. Full alignment could be replaced by “global alignment”.

Line 176: Review abbreviation of scientific names allover the text as well as species authors.

Line 186: Review grammar in sentence structure and the use of “as”.

Lines 157-160: Parameters for Sankey diagram are not clear. Looking the graphic in page 35, it is clearly possible to get more direct e simpler relationships. Review the graphic.

Line 277: Remove “Sanger sequencing”. Do the authors intend to say “Blast analyses”? This could be a suggestion.

Line 281: Please, include “e.g.” within the parentheses or explain why only those three targets were proposed as a solution.

## AUTHORS’ RESPONSE TO THE REVIEWERS

December 3, 2025

Manuscript ID: MIOC-2025-0260

Beyond *Leishmania*: Hidden trypanosomatid diversity reveals complex parasite-sand fly networks in Southeastern Brazil

Dear Dr. Cupolillo,

We sincerely thank you and the reviewers for the thorough and constructive evaluation of our manuscript. We carefully considered all comments and incorporated the suggested revisions to improve clarity, methodological transparency, and the overall scientific quality of the work. In several instances, the reviewers raised important conceptual and methodological points that helped us refine the presentation of our results and strengthen key interpretations.

Below, we provide a detailed, point-by-point response to all comments. Reviewer remarks are presented in italics, followed by our responses. For clarity, we refer to page and line numbers in the track change version of the manuscript. Whenever we slightly disagreed with a suggestion, we explain our reasoning while still ensuring that the final version addresses the reviewer’s concern.

We hope that the revised manuscript meets the expectations of the editorial board and reviewers, and we remain at your disposal for any additional adjustments needed.

Sincerely,

Felipe Dutra Rêgo

Instituto René Rachou - Oswaldo Cruz Foundation

**Reviewer comments to the authors**

**Reviewer #1:**

*General comment:*

*Dear Authors, the manuscript “Beyond Leishmania: Hidden trypanosomatid diversity reveals complex parasite-sand fly networks in Southeastern Brazil” (MIOC-2025-0260) fits very well within the scope of Memórias do Instituto Oswaldo Cruz.*

*Specific comments:*

*1. Include a discussion about the absence of a blood source analysis in engorged sand flies; However, it is understandable that there is not enough time to a blood source evaluation in the submitted manuscript.*

Authors’ comment: Thank you for your observation. No engorged females were collected during the sampling campaigns, which made blood-meal source analyses unfeasible in this study. We have now clarified this point in the Result section (lines 191-192) to avoid any impression of a methodological gap. Given the absence of engorged specimens, we believe that a more detailed discussion on this topic would fall outside the scope of the present work.

*2. Reduce the number of tables in the main text to two or three, condensing the results (see examples in: https://doi.org/10.1590/0074-02760200157);*

Authors’ comment: In accordance with your recommendation, we have reduced the number of tables presented in the main text. The tables containing the monthly distribution of sand fly species across the sampling campaigns and in peridomestic settings (former Table 2 and 5) have been moved to the Supplementary Material, as it provides detailed temporal information that is not essential for the interpretation of the main ecological and molecular findings.

*3. Give more attention to Crithidia, the most frequently detected genus in trypanosomatid surveys and recently associated with human disease (https://doi.org/10.3201/eid2511.181548).*

Authors’ comment: Thank you for this suggestion. However, no sequences obtained in our study clustered with *Crithidia* spp. in either BLAST searches or phylogenetic analyses. For this reason, a dedicated discussion on *Crithidia* would be speculative and not directly supported by our results. We therefore limited the discussion to the parasite groups detected in the study area, ensuring accuracy and coherence with the molecular identifications.

*Some small points must be solved before the final acceptance:*

*1. Keywords: Keywords should offer alternative ways for readers to find studies on similar topics. Please consider adding keywords that do not overlap with those in the title.*

Authors’ comment: We have revised the keywords to include broader and more searchable terms, improving discoverability and alignment with commonly indexed descriptors. The updated keywords are: Phlebotominae; *Leishmania*; Trypanosomatidae; Phylogenetic analysis; Vector-parasite interactions.

*2. Abstract (line 27): I believe “genera” fits better than “subfamilies”, but this is just a suggestion.*

Authors’ comment: The sentence has been updated accordingly (line 28).

*3. References: In-text citations should be represented by numbers corresponding to the reference list, which must be arranged in the order of citation. Please, adjust to the journal’s formatting guidelines.*

Authors’ comment: We have reformatted all in-text citations to the numerical style required by the journal and reorganized the reference list according to the order of appearance in the manuscript. The reference section now fully complies with the formatting guidelines of Memórias do Instituto Oswaldo Cruz.

*4. Lines 50–51: The mention “in targeted surveillance of Leishmania” reads more naturally than “Leishmania-targeted.”*

Authors’ comment: The phrasing has been amended accordingly, and “in targeted surveillance of *Leishmania*” has replaced “*Leishmania*-targeted” in the revised manuscript (line 47-50).

*5. Line 83: Is a single Shannon trap collection in July sufficient for this type of survey? Since different sand fly species were captured in the two traps, additional collections in other months could better represent sand fly diversity, don’t you think?*

Authors’ comment: We agree that additional Shannon trap collections could potentially capture further species and contribute to a more comprehensive faunal assessment. However, in this study the Shannon trap was used as a complementary method to obtain live females for dissection rather than as a primary tool for estimating species richness. Importantly, rarefaction and extrapolation analyses indicated that species richness reached asymptote, with observed and estimated richness (*Chao1*) coinciding and confidence intervals overlapping. This demonstrates that the sampling effort based on CDC traps was (apparently) sufficient to capture local diversity patterns, even with only one Shannon collection.

*6. Line 89: Similarly, the peridomestic survey may underestimate sand fly diversity.*

Authors’ comment: We agree that the peridomestic survey, conducted during a single sampling campaign, does not aim to represent the full diversity of sand flies in peri-urban environments of the Serra do Cipó region. The core objective of the study was the multi-campaign ecological and parasitological assessment of the Mata da Tapera forest fragment, and the peridomestic sampling served as a complementary component to explore parasite-vector interactions at the forest-household interface. The reduced sampling effort may underestimate local species richness; however, this limitation does not affect the main conclusions of the study, which rely on the more comprehensive dataset from the forest fragment. We have clarified this point in the revised manuscript (line 227-228). Importantly, expanded peridomestic surveys are currently underway in the region, aiming to generate a more complete inventory of sand fly fauna and to refine the preliminary patterns presented here.

*7. Natural Infection Analysis: This could be performed using more sensitive methods, such as inoculation of gut fragments into NNN/Schneider’s medium or other media more efficient for isolating monoxenous trypanosomatids and Trypanosoma spp. Have you considered this approach?*

Authors’ comment: We agree that isolation of gut fragments in culture media such as NNN, LIT or Schneider’s medium can be useful for recovering certain trypanosomatids. However, in sand flies this approach often faces substantial limitations. The midgut frequently contains high bacterial loads, and despite the use of antibiotics, contamination frequently overgrows cultures before trypanosomatids can be detected. In addition, monoxenous lineages and some *Trypanosoma* spp. differ markedly in their ability to grow in vitro, leading to low and heterogeneous recovery rates. Because of these constraints, there is currently no standardized protocol for natural infection analysis in phlebotomine sand flies. Studies variably rely on midgut dissection alone, dissection combined with PCR, direct PCR of whole females, or attempted culture isolation, each with specific drawbacks.

*8. Line 106: Considering that this is a general survey for trypanosomatids, would molecular detection in males not also be of interest? Monoxenous trypanosomatids are believed to be transmitted among insects mainly through water, oviposition, predation or coprophagy, and thus are not exclusively associated with blood-feeding specimens. However, screening male sand flies would not provide evidence for vertebrate-associated transmission.*

Authors’ comment: Thank you for this interesting comment. We agree that screening male sand flies could provide valuable insights into the ecology of monoxenous trypanosomatids, a topic that remains largely unexplored in the literature. We recognize the potential of this approach and consider it a promising avenue for future investigations.

*9. Lines 115 and 121: The PCR reactions for COI and 18S rRNA are described very briefly. Please provide additional methodological details, including cycling conditions, temperatures, and reaction steps.*

Authors’ comment: The COI (lines 124-129) and 18S rRNA (lines 134-139) PCRs were performed following the original protocols cited in the manuscript. To improve clarity, we have now added the essential cycling parameters (initial denaturation, annealing and extension temperatures, number of cycles, and reaction volumes) while keeping the description concise to avoid unnecessary duplication of previously established methods.

*10. Line 245: A reference to “personal communication” may be more appropriate when discussing the presence of L. infantum in mammals from the same study area.*

Authors’ comment: The sentence has been revised to refer to this information as a personal communication in the updated manuscript (line 270).

*11. Line 278: Mentions of Figures 3 and 4 in the Discussion section are unnecessary.*

Authors’ comment: The unnecessary mentions of Figures 3 and 4 in the Discussion have been removed in the revised version of the manuscript.

*12. References (line 347): They should be numbered and listed in order of appearance in the text.*

Authors’ comment: As noted above, all in-text citations have been converted to numerical format and the reference list has been reordered to follow the sequence of appearance in the manuscript.

*13. Figure 1: The study area could be more clearly illustrated with a map, similar to the illustration used in your group’s 2024 publication (https://doi.org/10.1371/journal.pone.0302567) and other studies cited in the manuscript.*

Authors’ comment: Following your recommendation, Figure 1 has been updated to include a clearer map of the study area. The revised figure provides improved geographic context and visualization of the sampling sites.

**Reviewer #2:**

*General comment:*

*The study addresses the infection of sand flies by Leishmania among other trypanosomatids, providing new findings on the circulation of these parasites in phlebotomine, including new records of sand fly species in Minas Gerais.*

*Specific comments:*

**
*Abstract*
**

*Line 15-17: I suggest that the authors better describe the methods used.*

Authors’ comment: We have revised the sentence to provide a clearer description of the methodological approach in the abstract. The updated version (line 15-18) now specifies that sampling was followed by midgut dissection and molecular screening for trypanosomatids.

**
*Introduction*
**

*Line 53-57: Citations are required in these sentences.*

Authors’ comment: We have added appropriate citations to support these statements in the Introduction. The revised version now includes references discussing the influence of periurban environments and habitat fragmentation on sand fly ecology and on the circulation of both *Leishmania* and non-*Leishmania* trypanosomatids (line 51-55).

**
*Methods*
**

*Why did the authors not also submit the dissected females for PCR? Were any engorged females observed?*

Authors’ comment: The females selected for midgut dissection (n = 105) were examined exclusively for the presence of flagellates, while molecular screening was conducted on a separate set of ethanol-preserved females. Midgut dissection typically leaves very limited biological material for reliable DNA extraction, which often results in low yield and reduced amplification success. Although PCR-based methods are generally sensitive, there is no consistent evidence that performing PCR on dissected midguts improves detection rates compared to screening intact females preserved specifically for molecular analyses. For this reason, PCR was applied only to intact ethanol-preserved females, ensuring higher DNA integrity and minimizing sample loss. No engorged females were observed in any of the collections, which made blood-meal analysis or PCR-based identification of vertebrate hosts unfeasible. This point has been clarified in the revised manuscript (line 191-192).

**
*Results*
**

*Tables: Table 1 already represents the fauna results well, I would suggest that Table 2 could be supplementary, as well as Table 3 and Table 5.*

Authors’ comment: Thank you for this observation. Following your suggestion, the table containing the monthly distribution of sand flies (former Table 2), the table summarizing dissected females, and the table listing peridomestic species (former Table 5) have been moved to the Supplementary Materials 2 and 4, respectively. We retained only three tables in the main text, those summarizing species composition, dissected females, and molecular identifications of trypanosomatid-positive specimens, because they contain essential information for interpreting the main ecological and parasitological findings.

*Table 4: I suggest that authors include the percentage of identity and query cover found in genbank. In the tables, the authors report the finding of Sc. birali, a recently described species. I suggest commenting on this new record for the state of Minas Gerais in the results and discussion.*

Authors’ comment: We have now added the percentage identity and query coverage values obtained from GenBank to Table 3 to improve transparency in the molecular identifications. We also included a brief comment in the Discussion section highlighting the record of *Sciopemyia birali*, which represents a new occurrence for the state of Minas Gerais.

*Figure 2. I suggest that the authors revise Figure 2. A better analysis would be across environments, rather than collection months. For example, an analysis of the richness between the forest and peridomestic environments.*

Authors’ comment: We agree that comparing species richness between forest and peridomestic environments can be informative; however, such an analysis requires equivalent sampling effort across sites. In our study, five sampling campaigns were conducted in the Mata da Tapera, whereas only a single campaign was carried out in peridomestic areas. Because richness estimates obtained through rarefaction and extrapolation are highly sensitive to sampling effort, a direct comparison between the two environments would not be methodologically appropriate. For this reason, Figure 2 focuses exclusively on the Mata da Tapera dataset, where the multi-campaign structure allows proper accumulation, interpolation, and extrapolation analyses. This approach enables us to evaluate sampling completeness and to verify that species richness reached asymptote, supporting the robustness of the ecological inferences drawn from that site.

*Regarding phylogenetic analysis, wouldn’t it be interesting to include Porcisia sequences?*

Authors’ comment: We agree that *Porcisia*, and other groups like *Endotrypanum* and *Leishmania (Mundinia)* represent relevant lineages within Leishmaniinae. However, none of the sequences obtained in our study clustered near these groups in preliminary BLAST analyses, and the phylogenetic placement of our isolates was consistently resolved within other well-supported clades. Including additional reference sequences from these genera would therefore not modify the inferred relationships or improve the interpretation of our results, while substantially increasing the size and complexity of the tree. For these reasons, we retained a focused phylogenetic framework containing the taxa necessary to robustly position the sequences generated in this study.

**
*Discussion*
**

*Regarding the discussion section, I am satisfied with the text presented, however, the authors could also explore and discuss a little more the results of the species richness found, as well as the new finding of Sciopemyia birali in Minas Gerais.*

Authors’ comment: We expanded the Discussion by adding a concise paragraph that places our species richness findings in the context of comparable surveys in Minas Gerais (e.g., Baldim and Jaboticatubas) and highlights the adequacy of our sampling effort in the Mata da Tapera forest fragment (lines 258-263).

**Reviewer #3:**

*General comment:*

*Abstract was presented properly with a summary of context, methods, results and main conclusions of the study. The manuscript has original results obtained with consistent and well-established sampling and molecular techniques but bring an alternative view for neglected Trypanosomatidae in sand fly surveys. The findings on this purpose bring the relevance of the manuscript. Methodology is appropriated but suggestion on its presentation were made for enhancing the manuscript. Results and main findings are well discussed. Figures 2-4 may have higher resolution. Suggestions on figure 5 were made in specific comments.*

*Specific comments:*

*Line 16: remove “(MT)”*

Authors’ comment: The abbreviation “(MT)” has been removed from the Abstract as suggested.

*Lines 49-51: It is not clear whether a broader range of parasite vectors interactions hampers the interpretation on Leishmania targeted surveillance. Please review the statement.*

Authors’ comment: We have revised the sentence to clarify that the presence of non-*Leishmania* trypanosomatids does not hinder surveillance itself, but rather increases the complexity of interpreting molecular results when broad-range markers are used (lines 47-50).

*Line 52: Reference needed.*

Authors’ comment: A reference has been added to support this statement in the revised manuscript (lines 51-55).

*Lines 57-59: Review the sentence considering that, from an evolutionary standpoint, it is unlikely that a sand fly species would develop a novel ability or environmental adaptation, but rather that such traits would be inherent outcomes of its evolutionary history.*

Authors’ comment: Thank you for the observation. The sentence has been revised to avoid implying recent evolutionary changes in sand fly species. The updated version clarifies that these species are naturally suited to anthropized environments, and that the uncertainty concerns their role in the ecology of non-*Leishmania* trypanosomatids (line 55-61).

*Line 66: Consider the substitution of “to characterize” by “to explore” once the study design description is not clear in terms of time scale an area coverage, even though ecological analyses adequately explore sampling efforts.*

Authors’ comment: The term “to characterize” has been replaced with “to explore” in the revised manuscript (line 63).

*Line 70: Please, consider including the source of the information or excluding it from the text.*

Authors’ comment: The information regarding autochthonous cases of canine visceral leishmaniasis has been revised and is now cited as personal communication from the Municipal Health Department (lines 66-67).

*Lines 83-86: in line 85 - “were separated using Castro aspirator” - it could seem to be out of sense considering collections in Shannon traps. Please review the text considering both sentences.*

Authors’ comment: Thank you for noting this ambiguity. The sentence has been revised to clarify that sand flies from CDC traps were removed from the trap cages using a Castro aspirator, whereas those collected in the Shannon trap were aspirated directly from the capture cloth. The revised text now provides a clearer description of both procedures (line 89-91).

*Lines 86-88: For me, it was not clear whether dissected sand flies were included or not in the molecular experiments. Please review the sentence.*

Authors’ comment: We have revised the paragraph to clarify how sand flies were collected from CDC and Shannon traps and to explicitly state that dissected females were not included in the molecular analyses. The revised text now makes the sampling and specimen allocation procedures fully clear (line 91-94).

*Lines 73-93: Sampling efforts presented seem to have some limitations in terms of time scale regarding the gap between MT and peridomestic survey hampering comparisons among the two collection series. It is also not clear whether there is a rational of geographic (special or ecological) distribution of collection sites. Do the collection sites represent the Serra do Espinhaço or even the Mata da Tapera? It would be nice to see this in the text or maybe in a map presenting the characteristics of the area and the collection sites.*

Authors’ comment: We have clarified the rationale behind the sampling design by specifying that the Mata da Tapera was the primary ecological survey site, sampled across five campaigns, while the peridomestic collections represent a single preliminary survey intended to document species presence at the forest-household interface rather than to enable direct ecological comparisons. We also expanded the description of the geographic context and revised Figure 1 to clearly display the spatial distribution of sampling sites (line 71-88).

*Line 96: what is the Leishmaniasis Group?*

Authors’ comment: The reference to the “Leishmaniasis Group” has been removed, and the sentence now reads simply “Fiocruz Minas,” which provides clearer institutional identification (line 102).

*Line 106: Review the sentence: “Females not used for direct dissection were processed individually”.*

Authors’ comment: The sentence has been revised to clarify that all females not used for midgut examination were still dissected individually for taxonomic identification prior to molecular screening (Line 111-114).

*Line 112: Review “and extraction blanks”.*

Authors’ comment: The sentence has been revised for clarity, and the role of extraction blanks as negative controls is now more clearly stated (line 117-119).

*Line 117: Consider replacing “under” by “and”.*

Authors’ comment: The word “under” has been replaced with “and” in the revised manuscript.

*Line 118: Review the description of the NTC considering that it should be a tube with the same mix but, instead of a sample, a volume of negative solution should be added.*

Authors’ comment: The description of the NTC has been revised for clarity. The text now specifies that the no-template control consisted of the complete reaction mix in which sample DNA was replaced by nuclease-free water (line 127-129).

*Lines 111-119: Protocols presented do not guarantee the integrity of pathogens DNA. In line 114, consider including “integrity of sand fly DNA”.*

Authors’ comment: The text has been revised to clarify that the COI PCR was used to verify the integrity of sand fly DNA rather than the integrity of pathogen DNA (lines 120-121).

*Line 129: Remove citation “Rêgo et al. 2015”.*

Authors’ comment: The citation has been removed.

*Lines 129-131: what were the parameters for consensus preparation? (e.g. phred quality?) Were there any ambiguous sites? If so, how were they treated? Were the primer sequences removed?*

Authors’ comment: The description of the sequencing workflow has been expanded to include the criteria used for consensus assembly. (lines 144-149)

*Line 134: There is a miss concept in “were combined with homologous sequences”. Sstructure suggestion: “Sequences for phylogenetic analyses were mined from Genbank by Blast filtering for Leishmaniinae, Phytomonadinae, Strigomonadinae, and Trypanosomatinae, with Bodo saltans included as the outgroup”*

Authors’ comment: The sentence has been revised for clarity (lines 154-157).

*Line 138: Why V7V8 low quality sequences were not removed in previous steps (line 129)?*

*Line 139: What is the electropherogram inspection and how was it made? Consider explaining the steps. Full alignment could be replaced by “global alignment”.*

Authors’ comment: Thank you for the observation. We have revised the text to clarify the sequence-processing workflow. Low-quality regions were removed during the manual inspection of chromatograms prior to consensus generation, and trimAl was subsequently applied only to refine the final alignment. The revised paragraph reflects this sequence of steps (lines 158-165).

*Line 176: Review abbreviation of scientific names all over the text as well as species authors.*

Authors’ comment: The abbreviations used for sand fly genera (e.g., Pi. for Pintomyia) follow the nomenclatural recommendations proposed by Marcondes (2007). A reference to this convention has now been added to the Methods section, and the use of abbreviations and species authors has been checked for consistency throughout the manuscript.

*Line 186: Review grammar in sentence structure and the use of “as”.*

Authors’ comment: The sentence has been revised to improve clarity and grammar (line 203-205).

*Lines 157-160: Parameters for Sankey diagram are not clear. Looking the graphic in page 35, it is clearly possible to get more direct e simpler relationships. Review the graphic.*

Authors’ comment: Thank you for the suggestion. We revised the Sankey diagram to improve clarity and reduce line crossings. The figure was regenerated in Flourish using the “Reduce overlaps” sorting option, which provides more direct and visually intuitive vector-parasite links. The Methods section has been updated to describe these parameters (lines 175-177).

*Line 277: Remove “Sanger sequencing”. Do the authors intend to say “Blast analyses”? This could be a suggestion.*

Authors’ comment: The term “Sanger sequencing” has been replaced with “BLAST analysis” in the revised manuscript (lines 297-298).

*Line 281: Please, include “e.g.” within the parentheses or explain why only those three targets were proposed as a solution.*

Authors’ comment: The sentence has been revised to clarify that the listed markers are examples commonly used in trypanosomatid phylogenetics (line 302).

---

## [Reviewer Report · REVIEWERS COMMENTS]

## REVIEWER #1

The authors have adequately addressed all points and suggestions raised in the first review. Congratulations on the manuscript. One final consideration concerns the citation of a “personal communication” on line 275, which should be avoided or replaced by a reference to another author, if possible.

## AUTHORS’ RESPONSE TO THE REVIEWERS

Dear Dr. Cupolillo,

Thank you very much for your careful evaluation of our revised manuscript entitled “Beyond *Leishmania*: Hidden trypanosomatid diversity reveals complex parasite-sand fly networks in Southeastern Brazil” (Manuscript ID: MIOC-2025-0260.R1). We are grateful for the positive feedback from the reviewer and for the final recommendation.

As suggested, we have removed the citation of “personal communication” previously indicated on line 275, ensuring full compliance with the journal’s editorial standards. Since no additional technical or methodological adjustments were required, we believe that the manuscript now fully meets all scientific and editorial criteria for publication in Memórias do Instituto Oswaldo Cruz. We sincerely appreciate the opportunity to revise our work and thank you and the reviewers for the constructive and encouraging evaluation throughout the process.

Kind regards,

Felipe Dutra Rêgo

Instituto René Rachou - Fiocruz Minas